# Real-Time Video Generation with Pyramid Attention Broadcast

**Xuanlei Zhao**[1*], **Xiaolong Jin**[2*], **Kai Wang**[1*†], **Yang You**[1†]
[1]National University of Singapore  [2]Purdue University
Code: NUS-HPC-AI-Lab/VideoSys

## Abstract

We present Pyramid Attention Broadcast (PAB), a *real-time*, *high quality* and *training-free* approach for DiT-based video generation. Our method is founded on the observation that attention difference in the diffusion process exhibits a U-shaped pattern, indicating significant redundancy. We mitigate this by broadcasting attention outputs to subsequent steps in a pyramid style. It applies different broadcast strategies to each attention based on their variance for best efficiency. We further introduce broadcast sequence parallel for more efficient distributed inference. PAB demonstrates up to $10.5\times$ speedup across three models compared to baselines, achieving real-time generation for up to 720p videos. We anticipate that our simple yet effective method will serve as a robust baseline and facilitate future research and application for video generation.

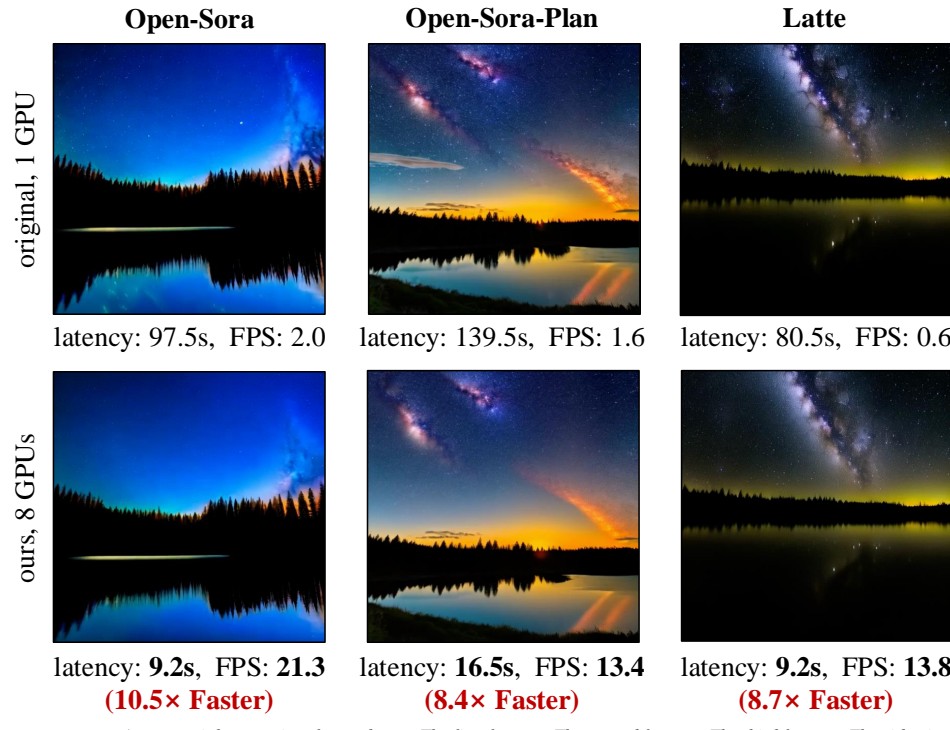

prompt: *A serene night scene in a forested area. The first frame ... The second frame ... The third frame ... The video is a time-lapse, capturing the transition from day to night, with the lake and forest serving as a constant backdrop. The style of the video is naturalistic, emphasizing the beauty of the night sky and the peacefulness of the forest.*

Figure 1: Results and speed comparison of our and original methods. PAB can significantly boost generation speed while preserving original quality. Latency is measured on 8 H100 GPUs. Video generation specifications: Open-Sora (2s, 480p), Open-Sora-Plan (2.7s , 512x512), Latte (2s, 512x512).

*equal contribution  †equal correspondence
{xuanlei, kai.wang, youy}@comp.nus.edu.sg  jin509@purdue.edu

# 1 INTRODUCTION

Sora (Brooks et al., 2024) kicks off the door of DiT-based video generation (Peebles & Xie, 2023). Recent approaches (Ma et al., 2024a; Zheng et al., 2024; Lab & etc., 2024) demonstrate their superiority compared to CNN-based methods (Blattmann et al., 2023; Wang et al., 2023a) especially in generated video quality. However, this improved quality comes from significant costs, *i.e.*, more memory occupancy, computation, and inference time. Therefore, exploring an efficient approach for DiT-based video generation becomes urgent for broader GenAI applications (Kumar & Kapoor, 2023; Othman, 2023; Meli et al., 2024).

Model compression methods employ techniques such as distillation (Crowley et al., 2018; Hsieh et al., 2023), pruning (Han et al., 2015; Ma et al., 2023), quantization (Banner et al., 2019; Lin et al., 2024), and novel architectures (Lin et al., 2024) to speedup deep learning models and have achieved remarkably success. Recently, they have also been proven to be effective on diffusion models (Sauer et al., 2023; Ma et al., 2024b; Chen et al., 2024b). Nevertheless, these methods usually require additional training with considerable computational resources and datasets, which makes model compression prohibitive and impractical especially for large-scale pre-trained models.

Most recently, researchers revisit the idea of cache (Smith, 1982; Goodman, 1983; Albonesi, 1999) to speedup diffusion models. Different from model compression methods, model caching methods are training-free. They alleviate redundancy by caching and reusing partial network outputs, thereby eliminating additional training. Some studies utilize high-level convolutional features for reusing purposes (Ma et al., 2024c) and efficient distributed inference (Li et al., 2024; Wang et al., 2024). Similar strategies have also been extended to specific attentions (Zhang et al., 2024; Wimbauer et al., 2024), *i.e.*, cross attention, and standard transformers (Chen et al., 2024c).

However, training-free speedup methods for DiT-based video generation still remains unexplored. Besides, previous model caching methods are not directly applicable to video DiTs due to two intrinsic differences: i) *Different architecture*. The model architecture has shifted from convolutional networks (Ronneberger et al., 2015) to transformers (Vaswani et al., 2017). This transaction makes former techniques that aims at convolutional networks not applicable to video generation anymore. ii) *Different components*. Video generation relies on three diverse attention mechanisms: spatial, temporal, and cross attention (Blattmann et al., 2023; Ma et al., 2024a). Such components lead to more complex dependency and attention interactions, making simple strategies ineffective. They also increase the time consumed by attentions, making attentions more critical than before.

To address these challenges, we take a closer look at attentions in video DiTs and empirically obtain two observations as shown in Figure 2: (i) The attention output differences between adjacent diffusion steps exhibit a U-shaped pattern, with stability in the middle 70% steps, indicating considerable redundancy for attention. (ii) Within the stable middle segment, different attention types also demonstrate various degrees of difference. Spatial attention changes the most with high-frequency visual elements, temporal attention shows mid-frequency variations related to movements, and cross-modal attention remains the most stable, linking text with video content (Zhang et al., 2024).

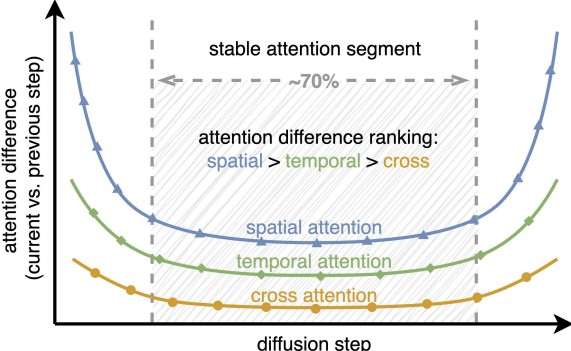

Figure 2: Comparison of the attention outputs differences between the current and previous diffusion steps. Differences are measured by Mean Square Error (MSE) and averaged across all layers for each diffusion step.

Based on these observations, we propose Pyramid Attention Broadcast (PAB), a *real-time*, *high quality* and *training-free* method for efficient DiT-based video generation. Our method mitigates attention redundancy by broadcasting the attention outputs to subsequent steps, thus eliminating attention computation in the diffusion process. Specifically, we apply various broadcast ranges for different attentions in a pyramid style, based on their stability and differences as shown in Figure 2. We empirically find that such broadcast strategy can also work to MLP layers. Additionally, to

enable efficient distributed inference, we propose broadcast sequence parallel, which significantly decreases generation time with much lower communication costs.

In summary, to the best of our knowledge, PAB is the first approach that achieves real-time video generation, reaching up to 35.6 FPS with a $10.5\times$ acceleration without compromising quality. It consistently delivers excellent and stable speedup across popular open-source video DiTs, including Open-Sora (Zheng et al., 2024), Open-Sora-Plan (Lab & etc., 2024), and Latte (Ma et al., 2024a). Notably, as a training-free and generalized approach, PAB has the potential to empower any future video DiTs with real-time capabilities.

## 2 HOW TO ACHIEVE REAL-TIME VIDEO GENERATION

### 2.1 PRELIMINARIES

**Denoising diffusion models.** Diffusion models are inspired by the physical process where particles spread out over time due to random motion, which consists of forward and reverse diffusion processes. The forward diffusion process gradually adds noise to the data over $T$ steps. Starting with data $\mathbf{x}_0$ from a distribution $q(\mathbf{x})$, noise is added at each step:

$$\mathbf{x}_t = \sqrt{\alpha_t}\mathbf{x}_{t-1} + \sqrt{1 - \alpha_t}\mathbf{z}_t \quad \text{for} \quad t = 1, \ldots, T, \tag{1}$$

where $\alpha_t$ controls the noise level and $\mathbf{z}_t \sim \mathcal{N}(0, \mathbf{I})$ is Gaussian noise. As $t$ increases, $\mathbf{x}_t$ becomes noisier, eventually approximating a normal distribution $\mathcal{N}(0, \mathbf{I})$ when $t = T$. The reverse diffusion process aims to recover the original data from the noisy version:

$$p_\theta(\mathbf{x}_{t-1}|\mathbf{x}_t) = \mathcal{N}(\mathbf{x}_{t-1}; \mu_\theta(\mathbf{x}_t, t), \Sigma_\theta(\mathbf{x}_t, t)), \tag{2}$$

where $\mu_\theta$ and $\Sigma_\theta$ are learned parameters defining the mean and covariance.

**Video generation models.** The remarkable success of Sora (Brooks et al., 2024) has demonstrated the great potential of diffusion transformers (DiT) (Peebles & Xie, 2023) in video generation, which leads to a series of research including Open-Sora (Zheng et al., 2024), Open-Sora-Plan (Lab & etc., 2024), and Latte (Ma et al., 2024a).

In this work, we focus on accelerating the DiT-based video generation models. As illustrated in Figure 3, we present the fundamental architecture of video DiTs. Different from transitional transformers, the model is composed of two types of transformer blocks: spatial and temporal. Spatial transformer blocks capture spatial information among tokens that share the same temporal index, while temporal transformer blocks handle information across different temporal dimensions. Cross-attention enables the model to incorporate information from the conditioning input at each step, ensuring that the generated output is coherent and aligned with the given context. Note that cross-attention mechanisms are not included in the temporal blocks of some video generation models (Ma et al., 2024a).

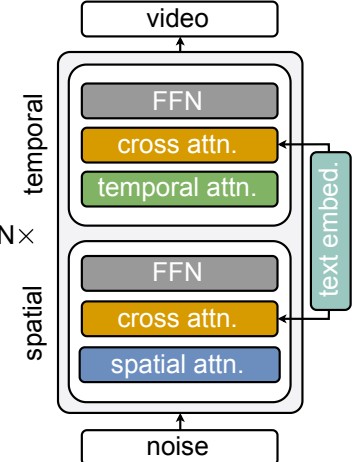

Figure 3: Overview of DiT-based video generation models, which compromises spatial and temporal transformer block. Cross attention incorporates information from text.

### 2.2 ATTENTION REDUNDANCY IN VIDEO DiTS

**Attention's rising costs.** Video DiTs employ three distinct types of attentions: spatial, temporal, and cross attention. Consequently, the computational cost of attention in these models is significantly higher than in previous methods. As Figure 4(b) illustrates, the proportion of time for total attention in video DiTs is significantly larger than in CNN approaches, which will further increases with larger video sizes. This dramatic increase poses a significant challenge to the efficiency of video generation.

**Unmasking attention patterns.** To accelerate costly attention components, we conduct an in-depth analysis of their behavior. Figure 4(a) shows the visualized differences in attention outputs across various stages. We observe that for middle segments, the differences are minimal and patterns appear

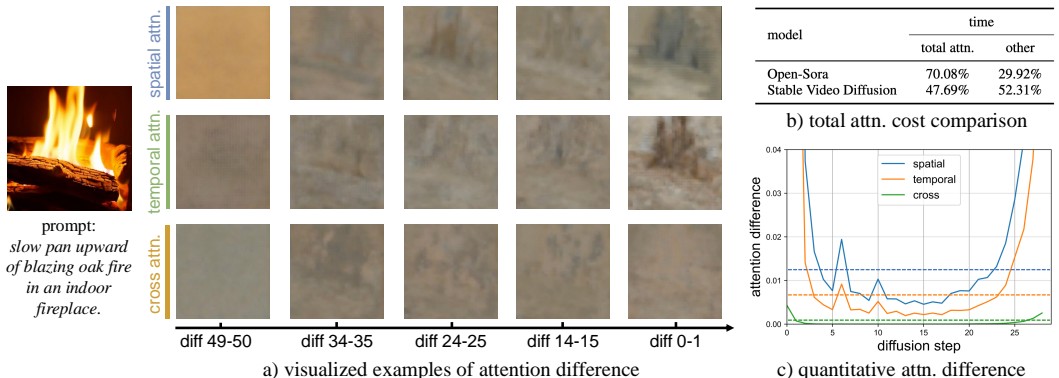

Figure 4: a) Visualization of attention differences in Latte. *diff i-j* represents the difference between step $i$ and step $j$. b) Comparison of total attention time cost between Stable Video Diffusion (Blattmann et al., 2023) (U-Net) and Open-Sora (DiT). c) Quantitative analysis of attention differences in Open-Sora, assessed using mean squared error (MSE). The dashed line represents the average value of the corresponding attention difference.

similar. The first few steps show vague patterns, likely due to the initial arrangement of content. In contrast, the final steps exhibit significant differences, presumably as the model sharpens features.

**Similarity and diversity.** To further investigate this phenomenon, we quantify the differences in attention outputs across all diffusion steps, as illustrated in Figure 4(c). Our analysis reveals that the differences in attention outputs demonstrate low difference for approximately 70% of the diffusion steps in the middle segment. Additionally, the variance in their outputs is also low, but still with slight differences: spatial attention shows the highest variance, followed by temporal and then cross-attention.

## 2.3 PYRAMID ATTENTION BROADCAST

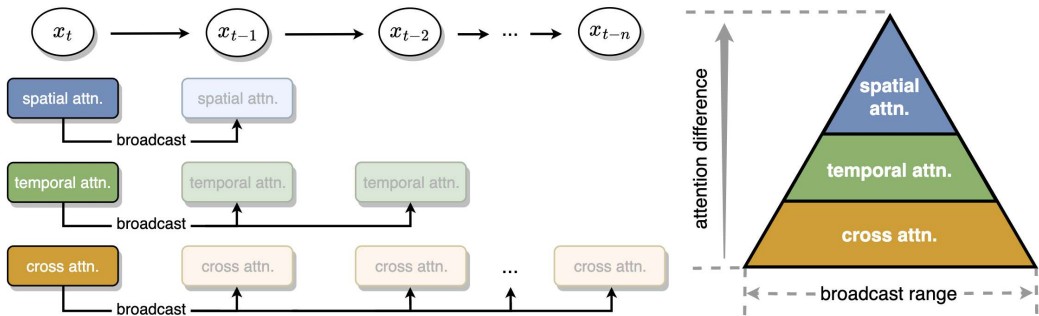

Figure 5: Overview of Pyramid Attention Broadcast. Our method (shown on the right side) which sets different broadcast ranges for three attentions based on their differences. The smaller the variation in attention, the longer the broadcast range. During runtime, we broadcast attention results to the next several steps (shown on the left side) to avoid redundant attention computations.

Building on the findings above, we propose Pyramid Attention Broadcast (PAB), a *real-time*, *high quality* and *training-free* method to speedup DiT-based video generation by alleviating redundancy in attention computations. As shown in Figure 5, PAB employs a simple yet effective strategy to broadcast the attention output from some diffusion steps to their subsequent steps within the stable middle segment of diffusion process. Different from previous approaches that reuse attention scores (Treviso et al., 2021), we choose to broadcast the entire attention module's outputs, as we find this method to be equally effective but significantly more efficient. This approach allows us to completely bypass redundant attention computations in those subsequent steps, thereby significantly reducing computational costs. This can be formulated as:

$$O_{\text{attn.}} = \{F(X_t), \underbrace{Y_t^*, \cdots, Y_t^*}_{\text{broadcast range}}, F(X_{t-n}), \underbrace{Y_{t-n}^*, \cdots, Y_{t-n}^*}_{\text{broadcast range}}, \cdots\}. \tag{3}$$

where $O_{\text{attn.}}$ refers to the output of the attention module at all timesteps, $F(X_t)$ denotes the attentions are calculated at timestep $t$ and $Y_t^*$ indicates the attentions results are broadcast from timestep $t$. We also apply similar strategy to mlp modules as depicted in Appendix A.2.2.

Furthermore, our research reveals that a single strategy across all attention types is still far from optimal, as each attention vary a lot as shown in Figure 2 and 4(c). To improve efficiency while preserving quality, we propose to tailor different broadcast ranges for each attention, as depicted in Figure 5. The determination of the broadcast ranges is based on two key factors: the rate of change and the stability of each attention type. Attention types that exhibit more changes and fluctuations at consecutive step are assigned smaller broadcast ranges for their outputs. This adaptive strategy enables more efficient handling of diverse attentions within the model architecture.

## 2.4 BROADCAST SEQUENCE PARALLELISM

We introduce broadcast sequence parallel, which leverages PAB's unique characteristics to improve distributed inference speed. Sequence parallel methods (Jacobs et al., 2023; Zhao et al., 2024) distributes workload across GPUs, thus reducing generation latency. But they incur significant communication overhead for temporal attention as shown in Figure 6. By broadcasting temporal attention, we

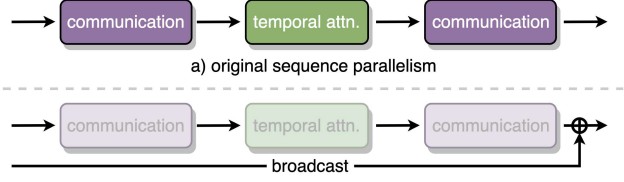

Figure 6: Comparison between original sequence parallelism and ours. When temporal attention is broadcast, we can avoid all communication.

naturally eliminate extra communications, substantially reducing overhead without quality loss, which enables more efficient, scalable distributed inference for real-time video generation.

## 3 EXPERIMENTS

In this section, we present our experimental settings, followed by our results and ablation studies. We then evaluate the scaling capabilities of our approach and visualize the results.

### 3.1 EXPERIMENTAL SETUP

**Models.** We select three state-of-the-art open-source DiT-based video generation models including Open-Sora-v1.2 (Zheng et al., 2024), Open-Sora-Plan-v1.1.0 (Lab & etc., 2024), and Latte-1.0 (Ma et al., 2024a) as our experimental models.

**Metrics.** Following previous works (Li et al., 2024; Ma et al., 2024a), we evaluate video quality using the following metrics: VBench (Huang et al., 2024), Peak Signal-to-Noise Ratio (PSNR), Learned Perceptual Image Patch Similarity (LPIPS) (Zhang et al., 2018), and Structural Similarity Index Measure (SSIM) (Wang & Bovik, 2002). VBench evaluates video generation quality, aligning with human perception. PSNR quantifies pixel-level fidelity between outputs, while LPIPS measures perceptual similarity, and SSIM assesses the structural similarity. The details of evaluation metrics are presented in Appendix A.4.

**Baselines.** We employ $\Delta$-DiT (Chen et al., 2024c) and T-GATE (Zhang et al., 2024), which are both training-free caching methods to accelerate DiTs. We show details in Appendix A.3.

**Implementation details.** All experiments are carried out on the NVIDIA H100 80GB GPUs with Pytorch. We enable FlashAttention (Dao et al., 2022) by default for all experiments.

| model | method | VBench ↑ | PSNR ↑ | LPIPS ↓ | SSIM ↑ | FLOPs (T) ↓ | latency (s) ↓ | speedup |
|---|---|---|---|---|---|---|---|---|
| Open-Sora | original | 79.22 | – | – | – | 3230.24 | 26.54 | – |
| | Δ-DiT | 78.21 | 11.91 | 0.5692 | 0.4811 | 3166.47 | 25.87 | 1.03× |
| | T-GATE | 77.61 | 15.50 | 0.3495 | 0.6760 | 2818.40 | 22.22 | 1.19× |
| | **PAB₂₄₆** | **78.51** | **27.04** | **0.0925** | **0.8847** | **2657.70** | 19.87 | 1.34× |
| | **PAB₃₅₇** | 77.64 | 24.50 | 0.1471 | 0.8405 | 2615.15 | 19.35 | 1.37× |
| | **PAB₅₇₉** | 76.95 | 23.58 | 0.1743 | 0.8220 | 2558.25 | **18.52** | **1.43×** |
| Open-Sora-Plan | original | 80.39 | – | – | – | 12032.40 | 46.49 | – |
| | Δ-DiT | 77.55 | 13.85 | 0.5388 | 0.3736 | 12027.72 | 46.08 | 1.01× |
| | T-GATE | 80.15 | 18.32 | 0.3066 | 0.6219 | 10663.32 | 39.37 | 1.18× |
| | **PAB₂₄₆** | **80.30** | **18.80** | **0.3059** | **0.6550** | **9276.57** | 33.83 | 1.37× |
| | **PAB₃₅₇** | 77.54 | 16.40 | 0.4490 | 0.5440 | 8899.32 | 31.61 | 1.47× |
| | **PAB₅₇₉** | 71.81 | 15.47 | 0.5499 | 0.4717 | 8551.26 | **29.50** | **1.58×** |
| Latte | original | 77.40 | – | – | – | 3439.47 | 11.18 | – |
| | Δ-DiT | 52.00 | 8.65 | 0.8513 | 0.1078 | 3437.33 | 10.85 | 1.02× |
| | T-GATE | 75.42 | 19.55 | 0.2612 | 0.6927 | 3059.02 | 9.88 | 1.13× |
| | **PAB₂₃₅** | **76.32** | **19.71** | **0.2699** | **0.7014** | **2767.22** | 8.91 | 1.25× |
| | **PAB₃₄₇** | 73.69 | 18.07 | 0.3517 | 0.6582 | 2648.45 | 8.45 | 1.32× |
| | **PAB₄₆₉** | 73.13 | 17.16 | 0.3903 | 0.6421 | 2576.77 | **8.21** | **1.36×** |

Table 1: Quality results on single GPU. $PAB_{\alpha\beta\gamma}$ denotes broadcast ranges of spatial ($\alpha$), temporal ($\beta$), and cross ($\gamma$) attentions. Video generation specifications: Open-Sora (2s, 480p), Open-Sora-Plan (2.7s, 512x512), Latte (2s, 512x512). PSNR, SSIM, and LPIPS are calculated against the original model results. *FLOPs* indicate floating-point operations per video generation.

## 3.2 MAIN RESULTS

**Quality results.** Table 1 presents quality comparisons between our method and baselines across four metrics and three models. We generate videos based on VBench's (Huang et al., 2024) prompts. Then evaluate VBench for each method, and calculate other metrics including PSNR, LPIPS, and SSIM with respect to the original results. $PAB_{\alpha\beta\gamma}$ denotes broadcast ranges of spatial ($\alpha$), temporal ($\beta$), and cross ($\gamma$) attentions. More experiments on other datasets can be found in Appendix B.1.

Based on the results, we make the following observations: i) Our method achieves superior quality results compared with two baselines while simultaneously achieving significantly higher acceleration by up to 1.58× on a single GPU. This demonstrates our method's ability to improve efficiency with negligible quality loss. ii) Our method consistently performs well across all three models, which utilize diverse training strategies and noise schedulers, demonstrating its generalizability.

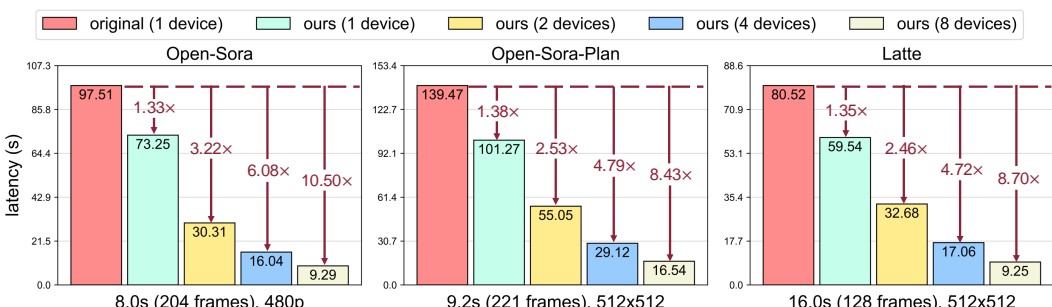

Figure 7: Speedups. We evaluate the latency and speedup achieved by PAB₂₄₆/PAB₂₃₅ (the strategy with best quality, but less speedup) for single video generation across up to 8 NVIDIA H100 GPUs. The results are presented for three models utilizing broadcast sequence parallelism. The multiple GPUs' speedup is compared with single GPU's speed.

**Speedups.** Figure 7 illustrates the significant speedup achieved by our method when leveraging multiple GPUs with broadcast sequence parallelism. Our method demonstrates almost linear speedups as the GPU number increases across three different models. Notably, it achieves an impressive 10.50×

speedup when utilizing 8 GPUs. These results highlight the significant reduction in communication overhead and underscore the efficacy of our broadcast sequence parallelism strategy.

## 3.3 ABLATION STUDY

To thoroughly examine the characteristics of our method, we conduct extensive ablation studies. Unless otherwise stated, we apply $PAB_{246}$ (the best quality, but less speedup) to Open-Sora for generating 2s 480p videos using a single NVIDIA H100 GPU.

Table 2: Evaluation of components. *w/o* indicates that the broadcast strategy is disabled only for the corresponding component.

| broadcast strategy | latency (s) ↓ | | VBench ↑ | |
|---|---|---|---|---|
| w/o spatial attn. | 21.74 | ↑1.87 | 78.45 | ↓0.06 |
| w/o temporal attn. | 23.95 | ↑4.08 | 78.98 | ↑0.47 |
| w/o cross attn. | 20.98 | ↑1.11 | 78.58 | ↑0.07 |
| w/o mlp | 20.27 | ↑0.40 | 78.59 | ↑0.08 |
| all components | 19.87 | – | 78.51 | – |

Table 3: Broadcast object comparison. We compare the speedup and effect for attention map and attention outputs. *attention outputs* refer to the final output of attention module. *attention scores* denotes attention score map.

| broadcast object | VBench ↑ | latency (s) ↓ |
|---|---|---|
| original | 79.22 | 26.54 |
| attention scores | **78.53** | 29.12 |
| attention outputs | 78.51 | **19.87** |

**Evaluation of components.** As shown in Table 2, we compare the contribution of each component in terms of speed and quality. We disable the broadcast strategy for each component individually and measure the VBench scores and increase in latency. While the impacts on VBench scores are negligible, all components contribute to the overall speedup. Spatial and temporal attentions yield the most computational savings, as they address more extensive redundancies compared to other components. Cross attention follows, offering moderate improvements despite its relatively lightweight computation. The mlp shows limited speedup due to its inherently low redundancy.

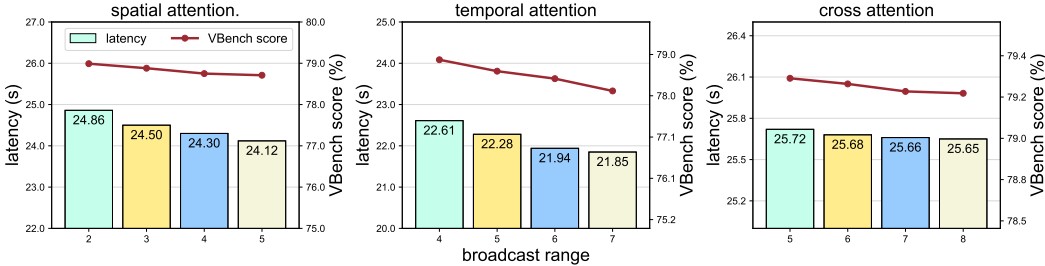

Figure 8: Evaluation of attention broadcast ranges. Comparison of latency and video quality across varying attention broadcast ranges in spatial, temporal, and cross attentions.

**Effect of attention broadcast range.** We conduct a comparative analysis of different broadcast ranges for spatial, temporal, and cross attentions. As illustrated in Figure 8, our findings reveal a clear inverse relationship between broadcast range and video quality. Moreover, we observe that the effect of different broadcast range varies across different attention, suggesting that each type of attention has its own distinct characteristics and requirements for optimal performance.

**What to broadcast in attention?** While previous works (Treviso et al., 2021) typically reuse attention scores, we find that broadcasting attention outputs is superior. Table 3 compares the speedup and video quality achieved by broadcasting attention scores versus attention outputs. Our results demonstrate that broadcasting attention outputs maintains similar quality while offering much better efficiency, for two primary reasons:

i) Attention output change rates are low, as the accumulated results after attention aggregation remain similar despite pixel-level changes. This further indicates significant redundancy in attention computations. ii) Broadcasting attention scores prevents the use of efficient attention kernels such as FlashAttention (Dao et al., 2022). It also requires complete attention-related computations, including attention calculation and linear projection, which are avoided when broadcasting outputs.

Table 4: Communication and latency of different sequence parallel methods on 8 NVIDIA H100 GPUs. *original* refers to our method on single GPU. *comm.* represents the total communication volume to generate a 8s 480p video.

| method | w/o PAB | | w/ PAB | |
|---|---|---|---|---|
| | comm. (G) | latency (s) | comm. (G) | latency (s) |
| original | – | 97.51 | – | 73.25 |
| Megatron-SP | 184.63 | 17.17 | 104.62 | 14.78 |
| DS-Ulysses | 46.16 | 12.34 | 26.16 | 9.85 |
| DSP | 23.08 | 12.01 | – | – |
| **ours** | – | – | **13.08** | **9.29** |

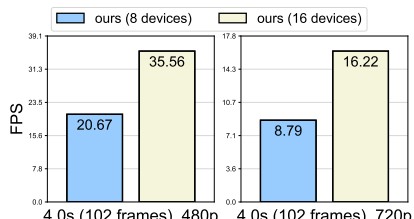

Figure 9: Real-time video generation performance. We evaluate our methods' speed in frames per second (FPS) using 8 and 16 NVIDIA H100 GPUs for 480p and 720p videos.

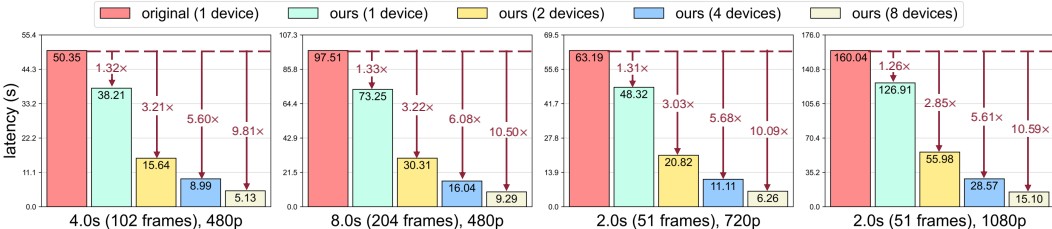

Figure 10: Scaling video size. Validating our method's acceleration and scaling capabilities on single and multi-GPU setups for generating larger videos.

## 3.4 SCALING ABILITY

To evaluate our method's scalability, we conduct a series of experiments. In each experiment, we apply $PAB_{246}$ (the best quality, but less speedup) to Open-Sora as our baseline configuration, change only the video sizes, parallel method and GPU numbers.

**Scaling to multiple GPUs.** We compare the scaling efficiency with and without our method using 8 GPUs in Table 4 for four sequence parallelism methods including Megatron-SP (Korthikanti et al., 2023), DS-Ulysses (Jacobs et al., 2023) and DSP (Zhao et al., 2024). Our broadcast sequence parallel is implemented based on DSP, and is also adaptable to other methods. The results demonstrate that: i) PAB significantly reduces communication volume for all sequence parallelism methods. Furthermore, our method achieves the lowest communication cost compared to other techniques, and achieving near-linear scaling on 8 GPUs. With a larger temporal broadcast range, it can yield even greater performance improvements. ii) Applying sequence parallelism alone is insufficient for optimal performance because of the significant communication overhead across multiple devices.

**Scaling to larger video size.** Currently, most models are limited to generating short, low-resolution videos. However, the ability to generate longer, higher-quality videos is both inevitable and necessary for future applications. To evaluate our model's capacity to accelerate processing for larger video sizes, we conducted tests across various video lengths and resolutions, as illustrated in Figure 10. Our results demonstrate that as video size increases, we can deliver stable speedup on a single GPU and better scaling capabilities when extending to multiple GPUs. These findings underscore the efficacy and potential of our method for processing larger video sizes.

**Real-time video generation.** We evaluate our method's speed in terms of FPS on 8 and 16 devices. Since in inference, the batch size of diffusion is often 2 because of CFG. Therefore, we split the batch first and apply sequence parallelism to each batch; in this way, PAB can extend to 16 devices with almost linear acceleration. As shown in Figure 9, we can achieve real-time with very high FPS video generation for 480p videos on 8 devices and even for 720p on 16 devices. Note that with acceleration techniques like Torch Compiler (Ansel et al., 2024), we are able to achieve even better speed.

**Runtime breakdown.** To further investigate how our method achieves such significant speedup, we provide a breakdown of the time consumption for various components, as shown in Figure 11. The analysis reveals that the attention calculation itself does not consume a large portion of time because the sequence length for attention will be much shorter if we do attention separately for each dimension.

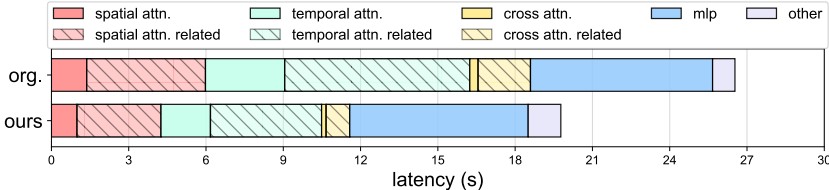

Figure 11: Runtime breakdown for generating a 2s 480p video. *attn.* denotes the time consumed by attention operations alone, while *attn. related* includes the time for additional operations associated with attention, such as normalization and projection.

However, attention-related operations, such as normalization and projection, are considerably more time-consuming than the attention mechanism itself, which mainly contribute to our speedup.

## 3.5 VISUALIZATION

As shown in Figure 12, we visualize the video results generated by our method compared to the original model. W we employ the highest quality strategy for each model. The visualized results demonstrate that our method maintains the original quality and details.

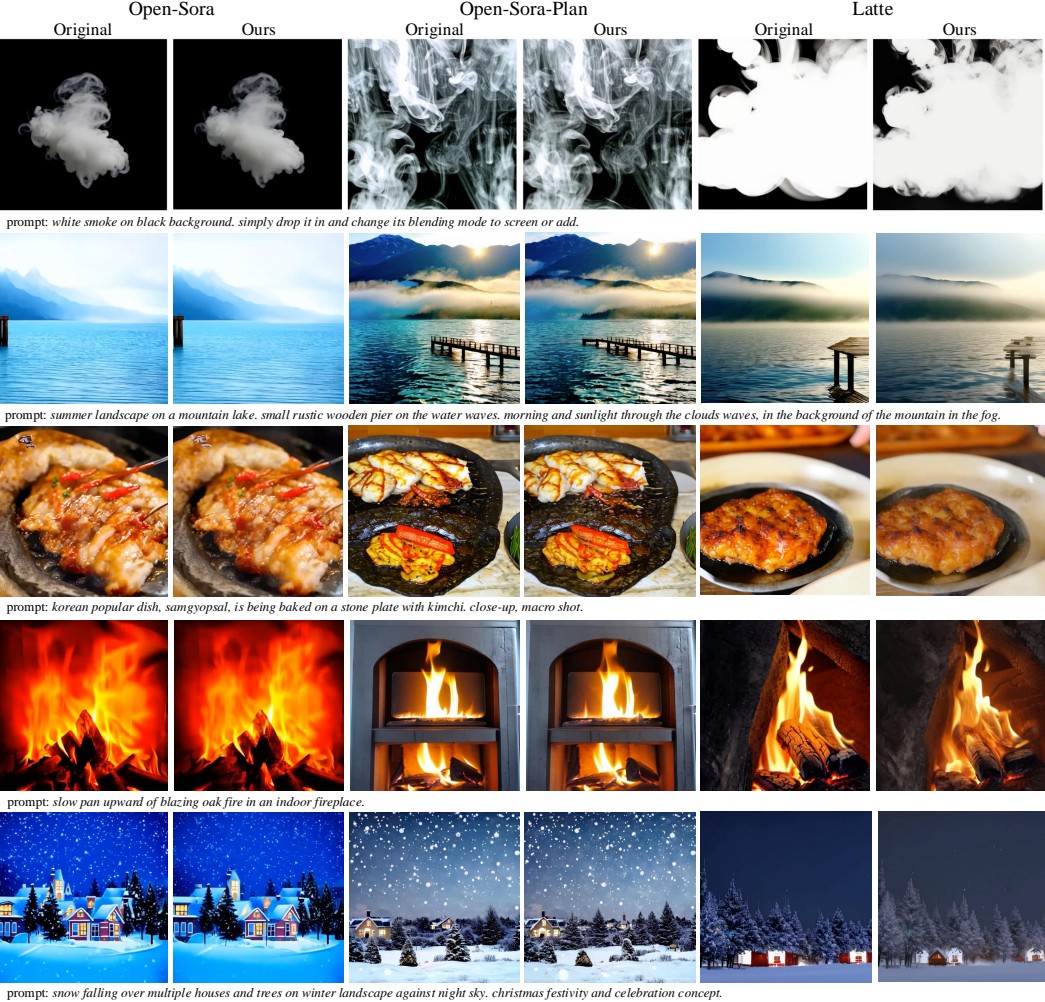

Figure 12: Qualitative results. We compare the generation quality between our method and original model. The figures are randomly sampled from the generated video.

## 4 RELATED WORK

### 4.1 VIDEO GENERATION

Early approaches of video generation primarily leveraged GANs (Goodfellow et al., 2014), VA-VAE (Van Den Oord et al., 2017), autoregressive Transformer (Rakhimov et al., 2020) and convolution-based diffusion models (Ho et al., 2022b). Recently Video generation has seen remarkable progress driven by diffusion models, which iteratively refine noisy inputs to generate high-fidelity video frames (Ho et al., 2022b; An et al., 2023; Esser et al., 2023; Chen et al., 2024a). While many works focus on conv-based diffusions (Ho et al., 2020; Harvey et al., 2022; Singer et al., 2022; Ho et al., 2022a; Luo et al., 2023; Wang et al., 2023b; Zhang et al., 2023) and achieve good results, researchers begin to explore Transformer-based diffusion models for video generation (Zheng et al., 2024; Lab & etc., 2024; Ma et al., 2024a) because of scalability and efficiency (Peebles & Xie, 2023).

### 4.2 DIFFUSION MODEL ACCELERATION

**Scheduler.** Reducing the sampling steps with schedulers has been explored through methods such as DDIM (Song et al., 2020), which enables fewer sampling steps without compromising generation quality. Other works also explore efficient solver of ODE or SDE (Song et al., 2021; Jolicoeur-Martineau et al., 2021; Lu et al., 2022; Karras et al., 2022; Lu et al., 2023).

**Compression.** Researchers aimed at reducing the workload and inference time at each sampling step, including distillation (Salimans & Ho, 2022; Li et al., 2023d), quantization (Li et al., 2023c; He et al., 2023; So et al., 2023a; Shang et al., 2023) and joint optimization (Li et al., 2023a; Liu et al., 2023). However, these methods demand significant resources and data for training, making them impractical especially for large-scale pre-trained models.

**Caching.** Recently, researchers revisited the concept of caching (Smith, 1982) in video generation for training-free acceleration. Some works (Ma et al., 2024c; Li et al., 2023b; Wimbauer et al., 2024; So et al., 2023b) reuse high-level features in U-Net structures while only updating low-level features based on the observation. However, these convolution-based methods can not directly apply to video DiTs. For transformer architectures, T-GATE (Zhang et al., 2024) introduce caching different attention at different stages, while $\Delta$-DiT (Chen et al., 2024c) propose to cache feature offsets of DiT blocks. Nevertheless, neither approach effectively addresses the unique attention features present in video DiTs, resulting in suboptimal performance.

**Parallelism.** Sequence parallelism techniques (Korthikanti et al., 2023; Jacobs et al., 2023; Zhao et al., 2024) have been proposed to reduce generation latency through distributed inference. However, these methods introduce additional communication costs. To address this issue, some works (Li et al., 2024; Wang et al., 2024) leverage convolutional features to reduce communication overhead. Nevertheless, these approaches are still limited to convolutions.

## 5 DISCUSSION AND CONCLUSION

In this work, we introduce Pyramid Attention Broadcast (PAB), a *real-time*, *high quality*, and *training-free* approach to enhance the efficiency of DiT-based video generation. PAB reduces attention redundancy through pyramid-style broadcasting by exploiting the U-shaped attention pattern in the diffusion process. Moreover, our broadcast sequence parallel significantly improves distributed inference efficiency. Overall, PAB achieves up to $10.5\times$ speedup with negligible quality loss and consistently outperforms baselines across various models. We believe that PAB provides a simple yet effective foundation for advancing future research and practical applications in video generation.

**Limitation.** PAB's performance may vary depending on the input data's complexity, especially with dynamic scenes. The fixed broadcast strategy might not work best for all video types and tasks. Also, we only focused on reducing redundancy in attention, not other parts of the model.

**Future works.** Our work opens several promising avenues for future research. One key direction is extending to more video models with diverse architectures. Another area is the redundancy in MLPs remains under-explored and warrants further investigation. Furthermore, our findings suggest potential for developing more efficient attention for video generation.

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

# Real-Time Video Generation with Pyramid Attention Broadcast

# Appendix

We organize our appendix as follows:

**Experimental Settings:**

**Additional Experimental Results and Findings:**

## A  EXPERIMENT SETTINGS

### A.1  MODELS

As we focus on DiT-based video generation, three popular state-of-the-art open-source DiT-based video generation models are selected in the evaluation, including Open-Sora (Zheng et al., 2024), Open-Sora-Plan (Lab & etc., 2024), and Latte (Ma et al., 2024a). Open-Sora-Plan (Lab & etc., 2024) utilizes CausalVideoVAE to compress visual representations and DiT with the 3D full attention module. Open-Sora (Zheng et al., 2024) combines 2D-VAE with 3D-VAE for better video compression and uses an SD-DiT block in the diffusion process. Latte (Ma et al., 2024a) uses spatial Transformer blocks and temporal Transformer blocks to capture video information in the diffusion process. The inference configs of three models are shown in Table 5, which strictly follow the official settings.

Table 5: The inference config of three models.

| model | scheduler | inference steps |
|---|---|---|
| Open-Sora | RFLOW | 30 |
| Open-Sora-Plan | PSNR | 150 |
| Latte | DDIM | 50 |

### A.2  PAB GENERATION SETTINGS

#### A.2.1  ATTENTION

In Table 6 we demonstrate the detailed settings of attention broadcast in experiments.

Table 6: The attention broadcast configuration of PAB. *diffusion timesteps* represents the start and end diffusion timestep of the broadcast, where 1000 is the beginning and 0 is the end.

| model | method | broadcast range | | | diffusion timesteps |
|---|---|---|---|---|---|
| | | spatial | temporal | cross | |
| Open-Sora | $PAB_{246}$ | 2 | 4 | 6 | [930-450] |
| | $PAB_{357}$ | 3 | 5 | 7 | |
| | $PAB_{579}$ | 5 | 7 | 9 | |
| Open-Sora-Plan | $PAB_{246}$ | 2 | 4 | 6 | [850-100] |
| | $PAB_{357}$ | 3 | 5 | 7 | |
| | $PAB_{579}$ | 5 | 7 | 9 | |
| Latte | $PAB_{235}$ | 2 | 3 | 5 | [800-100] |
| | $PAB_{347}$ | 3 | 4 | 7 | |
| | $PAB_{469}$ | 4 | 6 | 9 | |

### A.2.2 MLP

As demonstrated in Section 2.3 and Figure 4(c), the attention outputs exhibit low difference across approximately 70% of the diffusion steps within the middle segment. Spatial attention shows the highest variance, followed by temporal attention and, finally, cross-attention. Empirically, we perform a similar analysis on the MLP module to investigate whether it also involves redundant computations during the diffusion process.

In our current evaluation experiments, we select the skippable MLP modules for each model through empirical analysis in Appendix B.2. We show our detailed configuration for MLP modules in Table 7.

Table 7: The MLP broadcast configuration of PAB. *diffusion timesteps* represents the starting diffusion timestep of the broadcast and the *Block* indicates the index of the broadcast block.

| model | diffusion timesteps | block | broadcast range |
|---|---|---|---|
| Open-Sora | [864, 788, 676] | [0, 1, 2, 3, 4] | 2 |
| Open-Sora-Plan | [738, 714, 690, 666, 642, 618, 594, 570, 546, 522, 498, 474, 450, 426] | [0, 1, 2, 3, 4, 5, 6] | 2 |
| Latte | [720, 640, 560, 480, 400] | [0, 1, 2, 3, 4] | 2 |

### A.3 BASELINES GENERATION SETTINGS

We employ $\Delta$-DiT (Chen et al., 2024c) and T-GATE (Zhang et al., 2024), which are cache-based methods as baselines in the evaluation.

Table 8: Configuration of $\Delta$-DiT. $b$ represents the gate step of two stages and k is the cache interval. Block range refers to the index of the front blocks that are skipped. *Block range* refers to the specific indices of the blocks in the DiT-based video generation model that are skipped during the process. For example, *Block range* [0, 2] means that the first three blocks in the model block 0, block 1, and block 2—are skipped.

| $\Delta$-DiT | diffusion steps | b | k | block range |
|---|---|---|---|---|
| Open-Sora | 30 | 25 | 2 | [0, 5] |
| Open-Sora-Plan | 150 | 148 | 2 | [0, 1] |
| Latte | 50 | 48 | 2 | [0, 2] |

$\Delta$-DiT (Chen et al., 2024c) uses the offset of hidden states (the deviations between feature maps) rather than the feature maps themselves. $\Delta$-DiT is applied to the back blocks in the DiT during the early outline generation stage of the diffusion model and on front blocks during the detail generation stage. The stage is bounded by a hyperparameter $b$, and the cache interval is $k$. Since the source code for $\Delta$-DiT is not publicly available, we implement the baseline based on the methods in the paper. Additionally, we selected the parameters based on experimental results on video generation models.

We only jump the computation of the front blocks during the Outline Generation stage. The detailed configuration is shown in Table 8.

Table 9: Configuration of T-GATE. $m$ represents the gate step of the Semantics-Planning Phase and the Fidelity-Improving Phase, and $k$ is the cache interval.

| T-GATE | diffusion steps | m | k |
|---|---|---|---|
| Open-Sora | 30 | 12 | 2 |
| Open-Sora-Plan | 150 | 90 | 3 |
| Latte | 50 | 20 | 2 |

T-GATE (Zhang et al., 2024) reuses self-attention in semantics-planning phase and then skip cross-attention in the fidelity-improving phase. T-GATE segments the diffusion process into two phases: the semantics-planning phase and the fidelity-improving phase. Suppose $m$ represent the gate step of the transition between phases. Before gate step $m$, during the Semantics-Planning Phase, cross-attention (CA) remains active continuously, whereas self-attention (SA) is calculated and reused every $k$ steps following an initial warm-up period. After gate step $m$, cross-attention is replaced by a caching mechanism, with self-attention continuing to function. We present details in Table 9.

### A.4    METRICS

In this work, we evaluate our methods using several established metrics to comprehensively assess video quality and similarity. On the one hand, we assess video generation quality by the benchmark VBench, which is well aligned with human perceptions.

**VBench.** VBench (Huang et al., 2024) is a benchmark suite designed for evaluating video generative models, which uses a hierarchical approach to break down 'video generation quality' into various specific, well-defined dimensions. Specifically, VBench comprises 16 dimensions in video generation, including Subject Consistency, Background Consistency, Temporal Flickering, Motion Smoothness, Dynamic Degree, Aesthetic Quality, Imaging Quality, Object Class, Multiple Objects, Human Action, Color, Spatial Relationship, Scene, Appearance Style, Temporal Style, Overall Consistency. In experiments, we adopt the VBench evaluation framework and utilize the official code to apply weighted scores to assess generation quality.

On the other hand, we also evaluate the performance of the accelerated video generation model by the following metrics. We compare the generated videos from the original model (used as the baseline) with those from the accelerated model. The metrics are computed on each frame of the video and then averaged over all frames to provide a comprehensive assessment.

**Peak Signal-to-Noise Ratio (PSNR)**. PSNR is a widely used metric for measuring the quality of reconstruction in image processing. It is defined as:

$$\text{PSNR} = 10 \cdot \log_{10}\left(\frac{R^2}{\text{MSE}}\right), \tag{4}$$

where $R$ is the maximum possible pixel value of the image and MSE denotes the Mean Squared Error between the reference image and the reconstructed image. Higher PSNR values indicate better quality, as they reflect a lower error between the compared images. For video evaluation, PSNR is computed for each frame and the results are averaged to obtain the overall PSNR for the video. However, PSNR primarily measures pixel-wise fidelity and may not always align with perceived image quality.

**Learned Perceptual Image Patch Similarity (LPIPS).** LPIPS (Zhang et al., 2018) is a metric designed to capture perceptual similarity between images more effectively than pixel-based measures. It is based on deep learning models that learn to predict perceptual similarity by training on large datasets. It measures the distance between features extracted from pre-trained deep networks. The LPIPS score is computed as:

$$\text{LPIPS} = \sum_i \alpha_i \cdot \text{Dist}(F_i(I_1), F_i(I_2)), \tag{5}$$

where $F_i$ represents the feature maps from different layers of the network, $I_1$ and $I_2$ are the images being compared, Dist is a distance function (often L2 norm), and $\alpha_i$ are weights for each feature layer. Lower LPIPS values indicate higher perceptual similarity between the images, aligning better with human visual perception compared to PSNR. LPIPS is calculated for each frame of the video and averaged across all frames to produce a final score.

**Structural Similarity Index Measure (SSIM)**. SSIM (Wang & Bovik, 2002) evaluate the similarity between two images by considering changes in structural information, luminance, and contrast. SSIM is computed as:

$$\text{SSIM}(x, y) = \frac{(2\mu_x\mu_y + C_1)(2\sigma_{xy} + C_2)}{(\mu_x^2 + \mu_y^2 + C_1)(\sigma_x^2 + \sigma_y^2 + C_2)}, \tag{6}$$

where $\mu_x$ and $\mu_y$ are the mean values of image patches, $\sigma_x^2$ and $\sigma_y^2$ are the variances, $\sigma_{xy}$ is the covariance, and $C_1$ and $C_2$ are constants to stabilize the division with weak denominators. SSIM values range from -1 to 1, with 1 indicating perfect structural similarity. It provides a measure of image quality that reflects structural and perceptual differences. For video evaluation, SSIM is calculated for each frame and then averaged over all frames to provide an overall similarity measure.

## B    ADDITIONAL EXPERIMENTAL RESULTS AND FINDINGS

### B.1    ADDITIONAL QUANTITATIVE RESULTS.

| model | method | PSNR ↑ | | LPIPS ↓ | | SSIM ↑ | |
|---|---|---|---|---|---|---|---|
| | | w/ g.t. | w/ org. | w/ g.t. | w/ org. | w/ g.t. | w/ org. |
| Open-Sora | original | 8.62 | – | 0.7582 | – | 0.3506 | – |
| | $\Delta$-DiT | 9.44 | 12.01 | 0.7397 | 0.5263 | 0.3387 | 0.4676 |
| | T-GATE | 8.38 | 14.22 | 0.7658 | 0.3951 | 0.3811 | 0.6286 |
| | **PAB$_{246}$** | **8.69** | **26.53** | **0.7652** | **0.1001** | **0.3606** | **0.8635** |
| | **PAB$_{357}$** | 8.79 | 24.12 | 0.7719 | 0.1597 | 0.3695 | 0.8133 |
| | **PAB$_{579}$** | 8.84 | 22.48 | 0.7821 | 0.2129 | 0.3741 | 0.7745 |
| Open-Sora-Plan | original | 8.32 | – | 0.7701 | – | 0.2619 | – |
| | $\Delta$-DiT | 7.88 | 12.26 | 0.7719 | 0.5572 | 0.1884 | 0.3865 |
| | T-GATE | 8.39 | 13.60 | 0.7734 | 0.4750 | 0.2436 | 0.4544 |
| | **PAB$_{246}$** | **8.65** | **19.84** | **0.7653** | **0.2575** | **0.2759** | **0.6847** |
| | **PAB$_{357}$** | 8.87 | 17.39 | 0.7637 | 0.3814 | 0.2766 | 0.5767 |
| | **PAB$_{579}$** | 9.20 | 16.06 | 0.7610 | 0.4905 | 0.3025 | 0.4831 |
| Latte | original | 8.83 | – | 0.7670 | – | 0.3008 | – |
| | $\Delta$-DiT | 7.09 | 9.64 | 0.8071 | 0.7787 | 0.0741 | 0.1567 |
| | T-GATE | 9.27 | 19.13 | 0.7655 | 0.2585 | 0.3202 | 0.6416 |
| | **PAB$_{235}$** | **9.94** | **19.18** | **0.7743** | **0.2667** | **0.3742** | **0.6461** |
| | **PAB$_{347}$** | 10.38 | 17.49 | 0.7775 | 0.3577 | 0.4032 | 0.5813 |
| | **PAB$_{469}$** | 10.60 | 16.76 | 0.7832 | 0.3934 | 0.4190 | 0.5619 |

Table 10: Quality results on webvid. Latency and speedup are calculated on one GPU. $PAB_{\alpha\beta\gamma}$ denotes broadcast ranges of spatial ($\alpha$), temporal ($\beta$), and cross ($\gamma$) attentions. Video generation specifications: Open-Sora (2s, 480p), Open-Sora-Plan (2.7s, 512x512), Latte (2s, 512x512). *w/ g.t.* indicates evaluating the metrics based on the ground-truth videos, while *w/ org.* means with the original methods' outputs.

In Section 3.2, we present results only based on Vbench prompts. To further evaluate the efficacy of our method, we expand our analysis using a subset of 1000 videos from WebVid (Bain et al., 2021), a large-scale text-video dataset sourced from stock footage websites. We apply PAB to this subset, assessing its performance across three models and four metrics. The results of this additional experimentation are summarized in Table 10.

## B.2 FINDINGS FOR MLP BROADCAST.

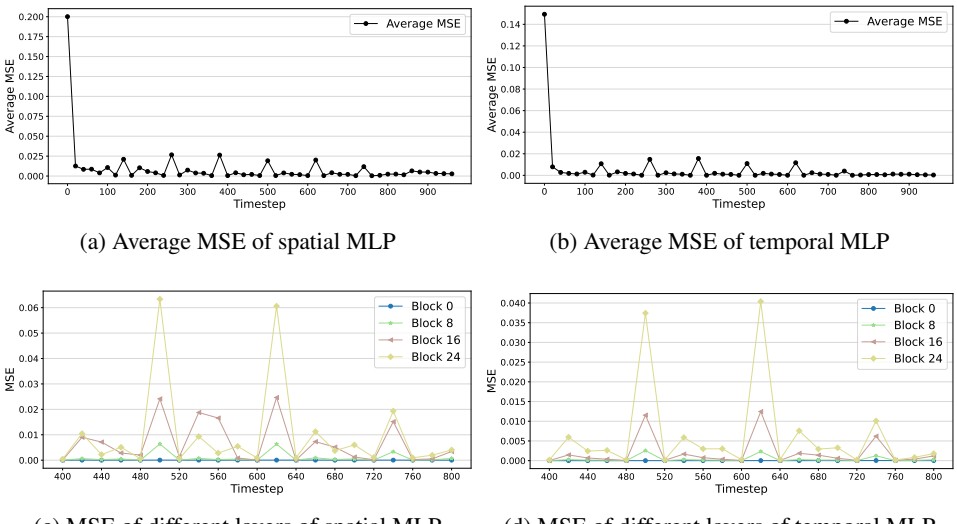

(a) Average MSE of spatial MLP

(b) Average MSE of temporal MLP

(c) MSE of different layers of spatial MLP

(d) MSE of different layers of temporal MLP

Figure 13: Quantitative analysis of MLP module differences in Latte by mean squared error (MSE) of the MLP output across continuous time steps. In Figures (a) and (c), we present the results for the spatial MLP, while Figures (b) and (d) illustrate the outcomes for the temporal MLP. Additionally, in Figure (a) and (b), we show the average MSE across all layers. In Figures (c) and (d), we select the block 0, 8, 16, and 24 to illustrate the characteristics of MLPs across different layers.

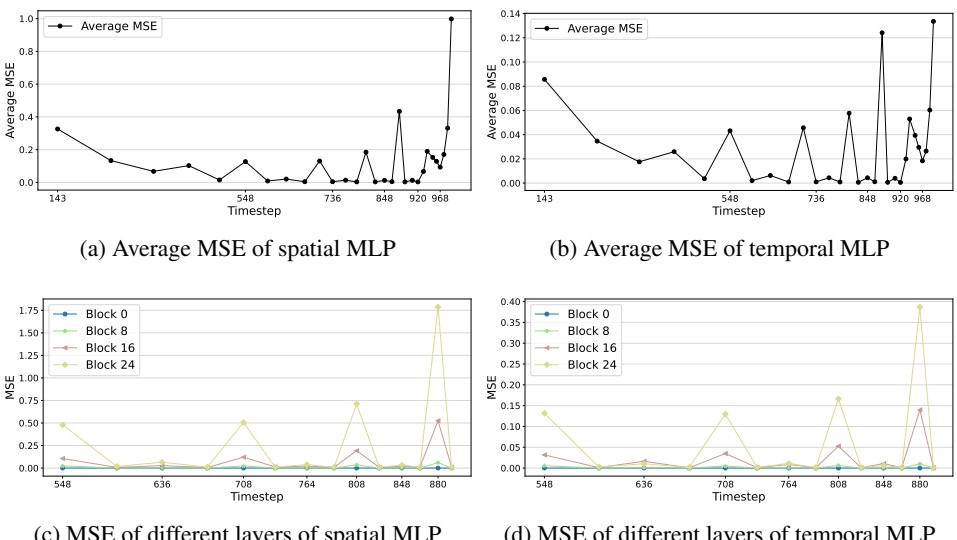

(a) Average MSE of spatial MLP

(b) Average MSE of temporal MLP

(c) MSE of different layers of spatial MLP

(d) MSE of different layers of temporal MLP

Figure 14: Quantitative analysis of MLP module differences in Open-Sora by mean squared error (MSE) of the MLP output across continuous time steps. Figures (a) and (c) present the results for the spatial MLP, while Figures (b) and (d) show the outcomes for the temporal MLP. Figures (a) and (b) display the average MSE across all layers, and Figures (c) and (d) examine block 0, 8, 16, and 24 to showcase the MLP characteristics across different layers.

We present a quantitative analysis of the FFN output differences in Latte, Open-Sora, and Open-Sora-Plan, using mean squared error (MSE) as the evaluation metric in Figure 13, 14 and 15.

We observe that during the intermediate stages of diffusion, the MSE exhibits a periodic spiking pattern, where local maxima occurs at specific timesteps, followed by consistently low values in subsequent timesteps. Therefore, we can retain the MLP output at the peak and reuse it during the

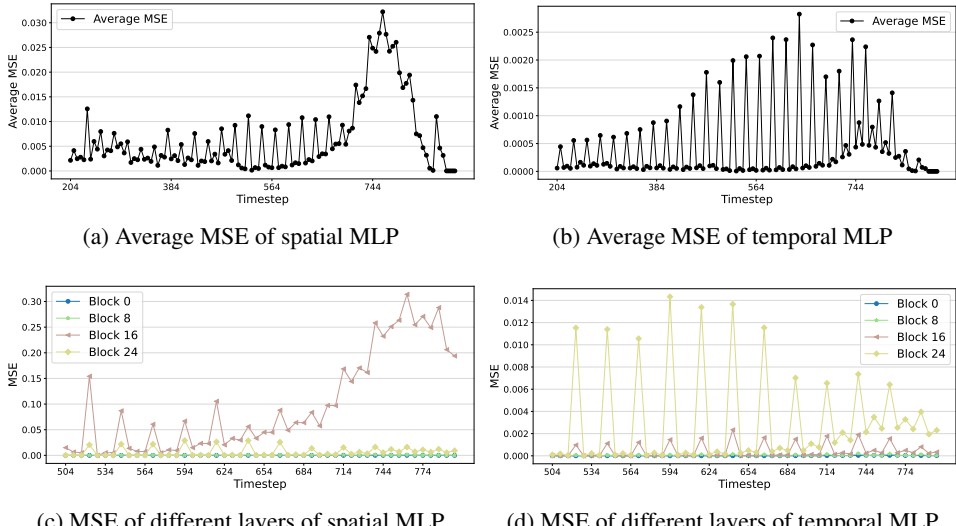

(a) Average MSE of spatial MLP  (b) Average MSE of temporal MLP

(c) MSE of different layers of spatial MLP  (d) MSE of different layers of temporal MLP

Figure 15: Quantitative analysis of MLP module differences in Open-Sora-Plan by mean squared error (MSE) of the MLP output across continuous time steps. In Figures (a) and (c), the results for the spatial MLP are shown, while Figures (b) and (d) show the results for the temporal MLP. Figures (a) and (b) display the average MSE across all layers, and Figures (c) and (d) highlight block 0, 8, 16, and 24 to illustrate the MLP behavior across different layers.

following low-value timesteps. Additionally, by analyzing the FFN modules across different blocks in Figure, we found that the output differences in the lower layers' MLPs are relatively small, while those in the upper layers' MLPs are significantly larger. Based on these findings, we empirically selected MLP modules to broadcast and corresponding broadcast ranges for each model, including Latte, Open-Sora, and Open-Sora-Plan.

### B.3 RESULTS FOR LONG, COMPLEX AND DYNAMIC SCENES.

In this section, we evaluate the quantitative and qualitative results for PAB when dealing with long, complex and dynamic scenes.

Table 11: Quantitative results of Open-Sora (16s 480p) on subset dimensions of Vbench datasets for long, complex and dynamic scenes.

| method | human action | overall consistency | imaging quality | aesthetic quality | dynamic degree | motion smoothness | subject consistency | total |
|---|---|---|---|---|---|---|---|---|
| original | 92.67 | 73.65 | 61.40 | 56.59 | 21.07 | 96.43 | 90.26 | 74.54 |
| $PAB_{246}$ | 91.33 | 73.43 | 60.18 | 56.24 | 19.91 | 97.05 | 90.08 | 73.98 |
| $PAB_{357}$ | 89.33 | 72.53 | 58.17 | 54.86 | 19.45 | 96.40 | 88.35 | 72.58 |
| $PAB_{579}$ | 88.33 | 72.36 | 57.92 | 54.63 | 18.05 | 96.50 | 88.32 | 72.15 |

**Quantitative results.** For model settings, we specifically use Open-Sora to generate videos of 16 seconds duration. This longer duration purposefully challenges our method with more complex and dynamic scenes. Open-Sora is the only model used as other models are restricted to fixed short lengths.

For dataset, from VBench's comprehensive 16-dimensional evaluation metrics, we strategically select 7 categories that best assess complex and dynamic scenes. Total performance scores are calculated based exclusively on these 7 categories to provide focused evaluation of complex and dynamic capabilities.

As shown in Table 11, we specifically test our method under the most challenging conditions by using the longest videos and selecting the more complex tasks in the dataset. The results show that PAB performs consistently well, with $PAB_{246}$ showing comparable performance.

What's particularly encouraging is that PAB maintains good scores even in the most difficult dimensions we tested, like human action and dynamic degree. This shows that our model stays reliable even under demanding conditions.

**Qualitative results.** As shown in Figure 16, our method demonstrates robust performance in processing dynamic, complex scenes while maintaining high-quality output.

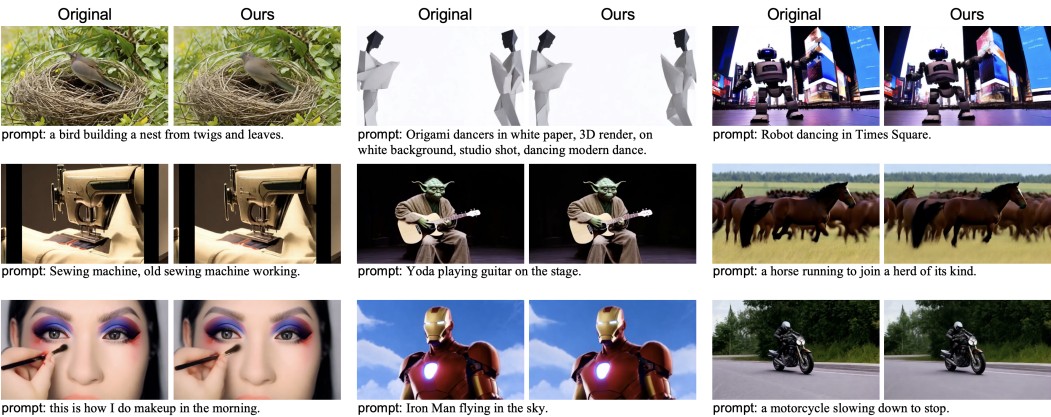

Figure 16: Qualitative results of Open-Sora (16s 480p) on subset dimensions of Vbench datasets for long, complex and dynamic scenes.

### B.4 BREAKDOWN OF PAB'S CONTRIBUTION WITH MULTIPLE GPUS

As shown in Table B.4, we evaluate the independent contribution of PAB from computation and communication and come with the following conclusions:

- With PAB's computation speedup (save computation by attention broadcast), the latency is further reduced by 24% compared with DSP only.
- With PAB's communication speedup (can save all communication cost when temporal attention is skipped), the latency can be further reduced by 5.0% compared with computation speedup only.
- Since we only evaluate based on $PAB_{246}$ (better quality but less speedup), PAB is able achieve more speedup if using more aggressive strategies.

Table 12: Breakdown of PAB's contribution with multiple GPUs using Open-Sora (8s 480p).

| method | latency (s) |
|---|---|
| original (1 gpu) | 96.90 |
| DSP (8 gpus) | 11.53 |
| DSP + PAB (with computation speedup) (8 gpus) | 9.29 |
| DSP + PAB (with computation and communication speedup) (8 gpus) | 8.85 |

### B.5 BREAKDOWN OF TIME COST WITHIN ATTENTION MODULE

As shown in Figure 11, the attention operation takes only a small proportion of time in attention module for 2s 480p Open-Sora. In this section, we further investigate what the main cost in attention module.

As shown in Table 13;14;15, our findings show that attention operation isn't actually the main thing slowing down the model. The real bottleneck comes from other parts - specifically layernorm and positional embedding. Even though these operations have fewer calculations and FLOPs, they run much slower in practice. Because modern GPUs are built to handle big matrix calculations super efficiently, but they struggle with operations that work on one element at a time, which is exactly what LayerNorm and positional embedding do.

Table 13: Breakdown of time cost in spatial attention.

| time | layernorm1 | mask | modulate | qkv proj | layernorm2 | o proj | attn | reshape |
|---|---|---|---|---|---|---|---|---|
| absolute (ms) | 1.132 | 0.149 | 0.616 | 0.473 | 2.160 | 0.176 | 1.595 | 0.312 |
| normalized | 17.1% | 2.2% | 9.3% | 7.2% | 32.7% | 2.7% | 24.1% | 4.7% |

Table 14: Breakdown of time cost in temporal attention.

| time | layernorm1 | mask | modulate | qkv proj | layernorm2 | pos emb | o proj | attn | reshape |
|---|---|---|---|---|---|---|---|---|---|
| absolute (ms) | 1.126 | 0.150 | 0.616 | 0.477 | 2.154 | 2.610 | 0.176 | 0.896 | 0.314 |
| normalized | 13.1% | 1.8% | 7.2% | 5.6% | 25.3% | 30.7% | 2.1% | 10.6% | 3.6% |

Table 15: Breakdown of time cost in cross attention.

| time | qkv proj | attn | o proj | reshape |
|---|---|---|---|---|
| absolute (ms) | 0.771 | 0.362 | 0.176 | 0.912 |
| normalized | 34.8% | 16.2% | 7.9% | 41.1% |

## B.6 WORKFLOW COMPARISON FOR BROADCASTING DIFFERENT OBJECTS

In Table 3, we demonstrate the efficiency of broadcasting different objects. In this section, we further demonstrate why there will be such difference:

- Broadcasting attention outputs enables us to bypass all intermediate computations within the attention module (including layer normalization, positional embedding, and qkvo projections) while maintaining compatibility with efficient attention kernels such as FlashAttention (we enable FlashAttention in all experiments by default to be closer to real-world usage).

- But broadcasting attention scores still requires partial computation in the attention module (e.g., attention calculation and linear projection). Its performance may even degrade below baseline due to incompatibility with FlashAttention.

To be more clear, here are the workflows in attention module for different broadcast strategies:

- original:

  $x \rightarrow q, k, v = proj(x) \rightarrow q, k = pos\_emb(layer\_norm(q, k)) \rightarrow o = attn(q, k, v) \rightarrow o = proj(o)$

- attention score (cannot use FlashAttention because we need attention score explicitly):

  $x \rightarrow v = proj(x) \rightarrow o = attn(broadcast\_score, v) \rightarrow o = proj(o)$

- attention outputs:

  $o = broadcast\_outputs$

## B.7 VARIOUS METRICS FOR EVALUATING REDUNDANCY

We evaluate different metrics for measuring redundancy as shown in Figure 17.

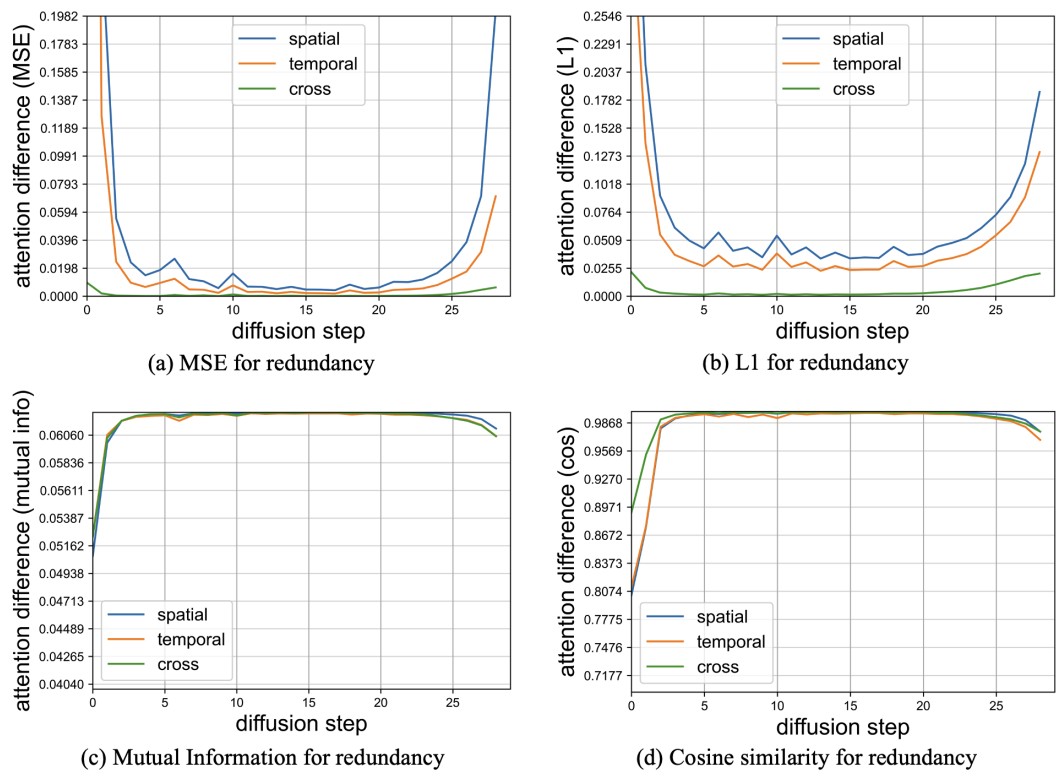

Figure 17: Various metrics for evaluating redundancy.

## B.8 EXTENSION TO TEXT-TO-IMAGE MODEL

PAB also has the potential to extend to Text-to-Image model like FLUX. In this section, we demonstrate our speedup, qualitative and quantitative results on FLUX.

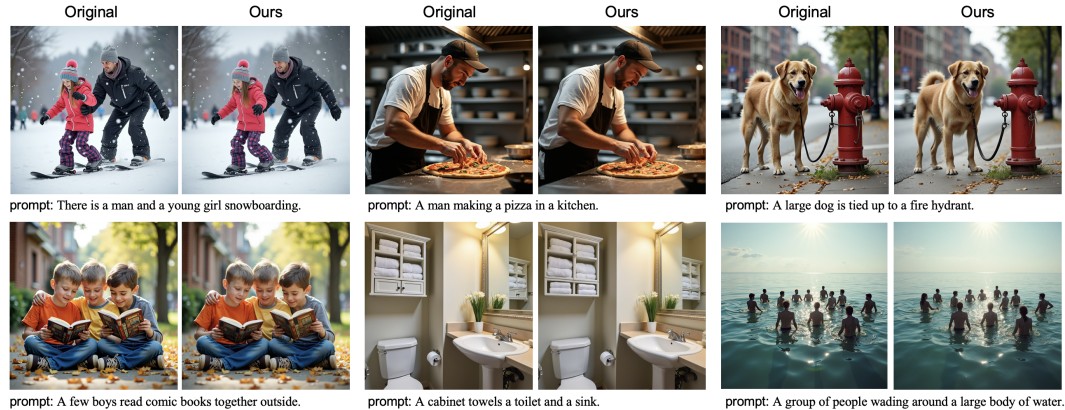

Figure 18: Qualitative results of FLUX with PAB$_{55}$.

Table 16: Speedup of FLUX. PAB$_{\alpha\gamma}$ denotes broadcast ranges of spatial ($\alpha$) and cross ($\gamma$) attentions.

| method | latency (s) |
|---|---|
| original | 13.8 |
| PAB$_{55}$ | 7.8 |

As shown in Table 16, we can achieve $1.77\times$ speedup compared with original method. We choose $PAB_{55}$ because it offers best balance between speedup and image quality. Note that this is only ran on single GPU.

As shown in Figure 18, we visualize the quantitative results on FLUX. Our method can achieve comparable results compared with baseline.

