# OpenReview forum: "Real-Time Video Generation with Pyramid Attention Broadcast"
_ICLR.cc/2025/Conference — ICLR 2025 Poster_

### Official Review · Reviewer_4GeG · 2024-10-30

**Soundness:** 3
**Presentation:** 3
**Contribution:** 3
**Rating:** 8
**Confidence:** 4

**Summary:**

This paper proposes an accelerated algorithm for video generation that is training-free. They observe the redundancy in attention calculations and adopts different acceleration strategies for different attention mechanisms. This paper gives detailed quantitative and qualitative results.

**Strengths:**

1)This paper is easy to follow and the code is open source.
2)he acceleration effect is obvious from the qualitative and quantitative experimental results,
3)The ablation experiment is thorough

**Weaknesses:**

1)This is essentially a cache technology, and the idea has been studied before, such as T-GATE and DeepCache.
2)The generalization of this method is questionable in scenarios with long video duration, complex textures, and drastic dynamic changes. For more details, please refer to Question 2 below.

**Questions:**

1)In Table 1, the PAB range is different for different models. Besides, the diffusion timesteps in the Appendix are different, such as Table 6 in the Appendix? Does this mean that these hyperparameters need to be determined experimentally? Will the hyperparameters used for the same model be very different when trained with different data?
 2)When generating long (greater than 1 minute), complex texture, high dynamic range videos, will acceleration lead to more quality loss. Can you give some quantitative results on how video quality metrics change as video length and complexity increase
 3)In Figure 11, why does the attention-related operation take up such a large proportion of the time? Are there other operations besides normalization and projection? Can you give a more detailed breakdown of the attention-related operations？

---

> ### Author Response · Authors · 2024-11-21
> **Response to reviewer 4GeG (1/4)**
>
> We sincerely thank the reviewer 4GeG for the valuable questions and comments. For the concerns and questions, here are our responses:
>
> **Q1: In Table 1, the PAB range is different for different models. Besides, the diffusion timesteps in the Appendix are different. Does this mean that these hyperparameters need to be determined experimentally?**
>
> **A1:** Thanks for the comment. Indeed the hyperparameters need to be determined experimentally, but it's very simple and easy.
>
> There are only two hyperparameters in PAB: 1) broadcast range. 2) the beginning and end steps of the broadcast.
>
> 1. **Broadcast range**
>
> **Experiment settings**: We conduct the following experiment to find out how to determine the best broadcast for all methods, where PAB$_{\alpha\beta\gamma}$ denotes broadcast ranges of spatial ($\alpha$), temporal ($\beta$), and cross ($\gamma$) attentions. The italicized results mean significant performance degradation.
>
> | method         | original | PAB$_{222}$ | PAB$_{235}$ | PAB$_{246}$ | PAB$_{257}$  | PAB$_{333}$  |
> | -------------- | -------- | ----------- | ----------- | ----------- | ------------ | ------------ |
> | Open-Sora      | 79.22    | 78.63       | 78.59       | **78.51**   | *77.88* | *78.07* |
> | Open-Sora-Plan | 80.39    | 80.33       | 80.33       | **80.30**   | *79.54* | *78.83* |
> | Latte          | 77.40    | 76.55       | **76.32**   | 76.25       | *76.02* | *75.86* |
>
> **Analysis**: As shown in the table, we can discover the following findings:
>
> * When we increase broadcast range from PAB$\_{222}$ to PAB$\_{246}$ or PAB$\_{235}$, there is not much performance degradation, as the redundancy degree of temporal and cross attention is larger. **So broadcast range should follow: cross range > temporal range > spatial range**.
> * When extending to PAB$\_{257}$ or PAB$\_{333}$, there will be severe performance degradation.
> * Although different models have different schedulers, timesteps, and training strategies, their best broadcast ranges are similar. **The optimal range is either PAB$\_{246}$ or PAB$\_{235}$** considering both effect and efficiency.
>
> 2. **The beginning and end steps of broadcast**
>
> **Experiment settings**: We conduct experiments to find out how to determine the best start and end steps for all methods, where $[\cdot, \cdot]$ denotes the end and start diffusion step of broadcast. The more steps it covers, the more speedup PAB can achieve. The italicized results mean significant performance degradation.
>
> | method         | original | [200, 700] | [150, 750] | [100, 800] | [100, 850] | [50, 950]    |
> | -------------- | -------- | ---------- | ---------- | ---------- | ---------- | ------------ |
> | Open-Sora      | 79.22    | 78.65      | 78.77      | **78.51**  | 78.20      | *77.21* |
> | Open-Sora-Plan | 80.39    | 80.30      | 80.22      | 80.28      | **80.30**  | *78.13* |
> | Latte          | 77.40    | 76.54      | 76.53      | **76.32**  | 75.90      | *75.66* |
>
> **Analysis**: As shown in the table, we can discover the following findings:
>
> * Reducing broadcast steps to $[150,750]$ or $[200,700]$ does not increase worthy performance, but leading to less speedup. However, increasing broadcast steps beyond $[50,950]$ leads to significant degradation.
> * Similarly, although different models have different schedulers, timesteps and training strategies, their best strategy is similar. **The optimal strategy is either $[100,800]$ or $[100,850]$**.
>
> **Conclusion: PAB is easy to find optimal hyperparameters.** It can consistently achieve good results for all three methods by setting:
>
> * **Broadcast range to PAB$\_{246}$ or PAB$\_{235}$**.
> * **Broadcast beginning and end steps to $[100,800]$ or $[100,850]$**.

---

> > ### Author Response · Authors · 2024-11-21
> > **Response to reviewer 4GeG (2/4)**
> >
> > **Q2: When generating long (greater than 1 minute), complex texture, high dynamic range videos, will acceleration lead to more quality loss. Can you give some quantitative results on how video quality metrics change as video length and complexity increase?**
> >
> > **A2:** We thank the comment. While the visualized examples focused on natural, static scenes, our method extends well beyond these cases.
> >
> > Our quantitative evaluation in Table 1 (Line 287) on the VBench dataset contains numerous examples of complex, dynamic actions, which partially demonstrate our method's broader effectiveness.
> >
> > To further clarify this, we conduct the following experiments on 16s video (the longest duration), dynamic and complex scenes:
> >
> > **For model settings**:
> >
> > * We specifically use Open-Sora to generate videos of **16 seconds duration** (the longest duration).
> > * This longer duration purposefully challenges our method with more complex and dynamic scenes.
> > * Open-Sora is the only model used as other models are restricted to fixed short lengths.
> >
> > **For dataset**:
> >
> > * From VBench's comprehensive 16-dimensional evaluation metrics, **we strategically select 7 categories that best assess complex and dynamic scenes**:
> >   * human action
> >   * overall consistency
> >   * imaging quality
> >   * aesthetic quality
> >   * dynamic degree
> >   * motion smoothness
> >   * subject consistency
> > * Total performance scores are averaged based exclusively on these 7 categories to provide evaluation of complex and dynamic capabilities.
> >
> > **Quantitative results**:
> >
> > | method      | human action | overall consistency | imaging quality | aesthetic quality | dynamic degree | motion smoothness | subject consistency | total |
> > | ----------- | ------------ | ------------------- | --------------- | ----------------- | -------------- | ----------------- | ------------------- | ----- |
> > | original    | 92.67        | 73.65               | 61.40           | 56.59             | 21.07          | 96.43             | 90.26               | 74.54 |
> > | PAB$_{246}$ | 91.33        | 73.43               | 60.18           | 56.24             | 19.91          | 97.05             | 90.08               | 73.98 |
> > | PAB$_{357}$ | 89.33        | 72.53               | 58.17           | 54.86             | 19.45          | 96.40             | 88.35               | 72.58 |
> > | PAB$_{579}$ | 88.33        | 72.36               | 57.92           | 54.63             | 18.05          | 96.50             | 88.32               | 72.15 |
> >
> > **Analysis**: As shown in table above, we specifically test our method under the most challenging conditions by using the longest videos and selecting the more complex tasks in the dataset. The results show that PAB performs consistently well, with PAB$_{246}$ achieving comparable performance compared with original method.
> >
> > What's particularly encouraging is that PAB maintains good scores even in the most difficult dimensions we tested, like human action and dynamic degree. This shows that our model stays reliable even under demanding conditions.
> >
> > **Qualitative results:** As shown in Figure 16 (Line XX), our method demonstrates robust performance in processing dynamic, complex scenes while maintaining high-quality output.
> >
> > Here are some example prompts:
> >
> > * Origami dancers in white paper, 3D render, on white background, studio shot, dancing modern dance.
> > * this is how i make up in the morning.
> > * Sewing machine, old sewing machine working.
> > * a horse running to join a herd of its kind.
> >
> > **Improvement plan**: In the revision, we will update both quantitative and qualitative results for PAB with long, complex, and dynamic scenes in appendix B.3 (Line 1015).
> >
> > **Conclusion: Both qualitative and quantitative results show our method is able to handle long, complex, and dynamic videos.**

---

> > > ### Author Response · Authors · 2024-11-21
> > > **Response to reviewer 4GeG (3/4)**
> > >
> > > **Q3: Why does the attention-related operation take up such a large proportion of the time?**
> > >
> > > **A3:** Here is the breakdown of attention related operations:
> > >
> > > * For spatial attention:
> > >
> > > | time          | layernorm1 | mask  | modulate | qkv proj | layernorm2 | o proj | attn  | reshape |
> > > | ------------- | ---------- | ----- | -------- | -------- | ---------- | ------ | ----- | ------- |
> > > | absolute (ms) | 1.132      | 0.149 | 0.616    | 0.473    | 2.160      | 0.176  | 1.595 | 0.312   |
> > > | normalized    | 17.1%      | 2.2%  | 9.3%     | 7.2%     | 32.7%      | 2.7%   | 24.1% | 4.7%    |
> > >
> > > * For temporal attention:
> > >
> > > | time          | layernorm1 | mask  | modulate | qkv proj | layernorm2 | pos emb | o proj | attn  | reshape |
> > > | ------------- | ---------- | ----- | -------- | -------- | ---------- | ------- | ------ | ----- | ------- |
> > > | absolute (ms) | 1.126      | 0.150 | 0.616    | 0.477    | 2.154      | 2.610   | 0.176  | 0.896 | 0.314   |
> > > | normalized    | 13.1%      | 1.8%  | 7.2%     | 5.6%     | 25.3%      | 30.7%   | 2.1%   | 10.6% | 3.6%    |
> > >
> > > * For cross attention:
> > >
> > > | time          | qkv proj | attn  | o proj | reshape |
> > > | ------------- | -------- | ----- | ------ | ------- |
> > > | absolute (ms) | 0.771    | 0.362 | 0.176  | 0.912   |
> > > | normalized    | 34.8%    | 16.2% | 7.9%   | 41.1%   |
> > >
> > > **Experiment settings**: We use Open-Sora v1.2.0 to generate a 2s 480p video on one NVIDIA H100 GPU. Time cost is shown in milliseconds within one layer.
> > >
> > > **Analysis**: Our findings show that attention operation isn't actually the main thing slowing down the model. The real bottleneck comes from other parts - specifically layernorm and positional embedding.
> > >
> > > Even though these operations have fewer calculations and FLOPs, they run much slower in practice. Because modern GPUs are built to handle big matrix calculations super efficiently, but they struggle with operations that work on one element at a time, which is exactly what LayerNorm and positional embedding do.
> > >
> > > **Improvement plan:** In the revision, we have updated the runtime breakdown for spatial, temporal and cross attention in the appendix section B.5 (Line 1131).
> > >
> > > **Conclusion: Operations like positional embedding and layernorm take most time in attention module.**

---

> > > > ### Author Response · Authors · 2024-11-21
> > > > **Response to reviewer 4GeG (4/4)**
> > > >
> > > > **Q4: This is essentially a cache technology, and the idea has been studied before.**
> > > >
> > > > **A4:** Thanks for the comment. PAB is indeed fundamentally a caching method. But we believe the significance of our work extends beyond this technical solution. Our paper presents two crucial empirical findings in video diffusion models:
> > > >
> > > > * There exists redundancy in attention for the middle 70% steps of diffusion.
> > > > * Different attentions have different redundancy degrees. Based on these two key findings, there can be many solutions to improve the efficiency of video models.
> > > >
> > > > Based on these two key findings, there are many methods to improve the efficiency of video generation models. For example:
> > > >
> > > > **Proposal 1:** We can make the model architecture more efficient by changing the number of each attention.
> > > >
> > > > **Motivation**: Now most models use equal amounts of spatial, temporal, and cross attention. However, our research shows that cross attention and temporal attention often do similar things (have high redundancy).
> > > >
> > > > **Method**: Therefore, we suggest changing the number of these attentions:
> > > >
> > > > * Spatial attention: every 1-2 layers (least redundant).
> > > > * Temporal attention: every 2-3 layers (less redundant).
> > > > * Cross attention: every 4-6 layers (most redundant).
> > > >
> > > > **Expected results & analysis**: This adjustment would help the model focus more on the content itself rather than spending too much effort analyzing relationships between elements. This could reduce redundant processing and potentially make the model more efficient.
> > > >
> > > > **Proposal 2:** We can make the diffusion scheduler more efficient by improving sample distribution.
> > > >
> > > > **Motivation**:  Our analysis revealed in Figure 1 that redundancy occurs in the middle 70% of the diffusion process. This insight suggests an opportunity to optimize the noise scheduler strategy by concentrating sampling efforts on the initial and final stages of diffusion, rather than the redundant middle segment.
> > > >
> > > > **Method**: We propose optimizing the noise scheduler by redistributing sampling density:
> > > >
> > > > * Allocate 60-70% of sampling steps to the critical initial (first 20%) and final (last 10%) stages.
> > > > * Reduce sampling ratio to 30-40% in the middle segments where redundancy is highest.
> > > >
> > > > **Expected results & analysis**: This approach could significantly improve both training efficiency and inference performance, and similar ideas have already been adopted by some latest works (https://arxiv.org/pdf/2410.13720, https://arxiv.org/pdf/2405.17403).
> > > >
> > > > **Conclusion:**
> > > >
> > > > * Our method is the first real-time video generation method with caching method, **but we view it as an initial step toward a broader goal: inspiring the development of more efficient video generation approaches.**
> > > > * These approaches include **improvements in model architecture** and **noise scheduler optimization** for both during training and inference.
> > > > * We envision our work contributing to **making video generation more practical and accessible for real-world applications**.
> > > >
> > > > We are particularly happy to continue the discussion in the discussion phase!

---

> > > > > ### Author Response · Authors · 2024-11-25
> > > > > **Looking forward to the reply**
> > > > >
> > > > > Dear reviewer 4GeG:
> > > > >
> > > > > Thanks so much again for the time and effort in our work. According to the comments and concerns, we conduct the corresponding experiments and further discuss the related points. Additionally, we have added all improvements including 6 detailed experiments in the latest pdf revision.
> > > > >
> > > > > As the rebuttal period is about to close, may I know if our rebuttal addresses the concerns? If there are further concerns or questions, we are welcome to address them. Thanks again for taking the time to review our work and provide insightful comments.

---

> > > > > > ### Comment · Reviewer_4GeG · 2024-11-29
> > > > > >
> > > > > > Thank you for your responses, most of my concerns are addressed. I will raise my rating.

---

> > > > > > > ### Author Response · Authors · 2024-11-29
> > > > > > >
> > > > > > > Dear reviewer 4GeG,
> > > > > > >
> > > > > > > We would like to express our sincere gratitude to reviewer 4GeG for acknowledging ourwork, providing constructive suggestions and raising the score.
> > > > > > >
> > > > > > > Thanks again for the time and effort in reviewing our work.

---

### Official Review · Reviewer_6TRe · 2024-11-01

**Soundness:** 4
**Presentation:** 4
**Contribution:** 3
**Rating:** 8
**Confidence:** 4

**Summary:**

This paper proposes Pyramid Attention Broadcast (PAB), which is a real-time, high-quality, and training-free approach for DiT-based video generation. Motivated by the observation that attention difference in the diffusion process exhibits a U-shaped pattern, indicating significant redundancy. This paper broadcast attention outputs to subsequent steps in a pyramid style. PAB demonstrates up to 10.5x speedup across three models compared with baselines, achieving real-time generation for 720p vdeos.

**Strengths:**

1. Figure 2 vividly illustrates the motivation, and the results appear highly plausible, with the spatial attention difference being the most significant, followed by the temporal attention difference, and then the cross attention difference.

2. The proposed broadcasting method appears to be straightforward, potent, and adaptable.

3. Broadcasting the entire sequence in parallel significantly enhances inference speed. It seems that sequence parallelism is exceptionally effective for accelerating the process.

**Weaknesses:**

* Why opt for the Mean Squared Error (MSE) metric to assess redundancy, and does the MSE metric accurately reflect the nuances of redundancy?

* It appears that utilizing broadcasting may lead to an increase in communication time. How can we quantify this additional overhead?

* Is it feasible to apply the PAB (Parallel Attention Broadcasting) technique to T2I (Text-to-Image) models, such as FLUX? Given that current T2I models are becoming increasingly large, there is a pressing need to accelerate models that exceed 5 billion parameters.

**Questions:**

Overall, I find this to be a well-crafted paper. Nevertheless, I am curious if there might be a more suitable metric for assessing redundancy. Additionally, I wonder if it's possible to dynamically select layer-wise broadcasting in conjunction with distillation-based methods to facilitate real-time video generation.

---

> ### Author Response · Authors · 2024-11-21
> **Response to reviewer 6TRe (1/2)**
>
> We sincerely thank the reviewer 6TRe for the valuable questions and comments. For the concerns and questions, here are our responses:
>
>
> **Q1: Why opt for the Mean Squared Error (MSE) metric to assess redundancy, and does the MSE metric accurately reflect the nuances of redundancy? If there might be a more suitable metric for assessing redundancy?**
>
> **A1:** Thanks for the comment.
>
> We choose Mean Squared Error to access redundancy because:
>
> * Outliner sensitivity: More sensitive to outliers than other losses like L1 loss (which means less redundancy), and tend to reduce the impact of really small changes (which means redundancy). Thus more sensitive to the change rate of latents, helping us to identify the redundant region.
> * Theoretical basis: Aligns with Gaussian noise in the diffusion process, matching the likelihood function of Gaussian distributions.
>
> There are also some metrics that can reflect redundancy, including MSE, L1, cosine similarity and mutual information, as shown in Figure 17 (Line 1188).
>
> We find all of the metrics indicates the similar redundancy region (the middle 70% region) as MSE. If you have any other suggestions for the metric, welcome to propose!
>
>
> **Q2: It appears that utilizing broadcasting may lead to an increase in communication time. How can we quantify this additional overhead?**
>
> **A2:** Simply applying broadcast will not increase communication cost, because all outputs is stored and broadcast on the local machine.
>
> But using multiple GPUs will increase communication cost, because it will need to change the sequence layout among GPUs for temporal attention. PAB can help to reduce the cost of such communication as shown in Figure 6 (Line 232), because PAB can skip the whole temporal attention module, thus eliminating the need for communication.
>
> To demonstrate how PAB contributes in multiple GPUs case. Here we demonstrate the breakdown the contribution of PAB.
>
> **Experiments**:
>
> | method                                                       | latency (s) |
> | ------------------------------------------------------------ | ----------- |
> | original (1 gpu)                                             | 96.90       |
> | DSP (8 gpus)                                                 | 11.53       |
> | DSP + PAB (with computation speedup) (8 gpus)                | 9.29        |
> | DSP + PAB (with computation and communication speedup) (8 gpus) | 8.85        |
>
> **Experiment settings**: DSP is the sequence parallel method we use. We follow the settings of Table 4, using Open-Sora to generate 8s of 480p videos. The speed is slightly different from the results in Table 1 because we are using different machines and environments.
>
> **Analysis**: As shown in the experiment, we evaluate the independent contribution of PAB from computation and communication.
>
> * With PAB's computation speedup (save computation by attention broadcast), the latency is further reduced by 24% compared with DSP only.
> * With PAB's communication speedup (can save all communication costs when temporal attention is skipped), the latency can be further reduced by 5.0% compared with computation speedup only.
> * Since we only evaluate based on PAB$_{246}$  (better quality but less speedup), PAB is able to achieve more speedup if using more aggressive strategies.
>
> **Conclusion: PAB can help to reduce both computation and communication costs.**
>
> **Q3: Is it feasible to apply PAB to T2I (Text-to-Image) models, such as FLUX?**
>
> **A3:** Yeah, although T2I models only have spatial attention and cross attention, PAB can also generalize well to these methods. We adapt PAB to FLUX by applying spatial and temporal attention broadcast only. PAB$_{\alpha\gamma}$ denotes broadcast ranges of spatial ($\alpha$) and cross ($\gamma$) attentions.
>
> **Speedup**: PAB can achieve **1.77x** speedup for FLUX with minor quality loss using PAB$_{55}$ . We choose it because it offers best balance between speedup and image quality.
>
> Note that this is only ran on single GPU. If you are interested in multiple performance with real-time performance, we are willing to implement!
>
> | method     | latency (s) |
> | ---------- | ----------- |
> | original   | 13.8        |
> | PAB$_{55}$ | 7.8         |
>
> **Qualitative results**: In appendix section B.8 (Line 1216), we update the qualitative results for FLUX with PAB$_{55}$. As shown in Figure 18 (Line 1221), the image quality is basically comparable with the original methods, which proves the generalization and effectiveness for PAB.
>
> **Quantitative results**: We are still runing experiments for quantitative results with more PAB settings, but it will take some time because the model is big. We will provide update as soon as we get the final results.
>
> **Improvement plan**: In the revision, we will update the quanlitative and quantitative results in the appendix B.8 (Line 1216). And the source code of PAB implemention for FLUX will be uploaded as well.

---

> ### Author Response · Authors · 2024-11-21
> **Response to reviewer 6TRe (2/2)**
>
> **Q4: If it's possible to dynamically select layer-wise broadcasting in conjunction with distillation-based methods to facilitate real-time video generation.**
>
> **A4:** Indeed, the integration of layer-wise broadcasting with distillation-based methods is technically feasible and potentially advantageous. The approach is particularly promising because distillation methods typically constrain generation to a fixed number of steps (e.g., 1 or 4), while our proposed method can be adapted to dynamically optimize redundancy reduction across different layers.
>
> Given layers and data exhibit varying degrees of redundancy and distinct computational characteristics, this combination could offer complementary benefits - distillation providing speed-up through step reduction, and PAB enabling fine-grained optimization of computation allocation.
>
> For example, a simple idea is to visualize the redundancy of attentions for each layer like in Figure 4c (Line 162), then apply approppriate broadcast strategy for them.
>
> However, due to the current absence of distillation-based video generation models, we cannot conduct experimental validation of this integration. We believe this represents a promising direction for future research that could further advance real-time video generation capabilities.
>
> We are particularly happy to continue the discussion in the discussion phase!

---

> > ### Author Response · Authors · 2024-11-25
> > **Update on quantitative results of FLUX**
> >
> > Here are the quantitative results of Flux on 5000 images of coco-eval datasets.
> >
> > | method |FID score (lower is better)|
> > |-----|----|
> > |original|36.05|
> > |PAB$\_{55}$|33.57|
> >
> > Analysis: PAB can achieve 1.77x speedup while mainatining comparable performance compared with original FLUX.

---

> > > ### Comment · Reviewer_6TRe · 2024-11-29
> > > **To authors**
> > >
> > > Thanks for your response.
> > >
> > > I keep my score.

---

> > > > ### Author Response · Authors · 2024-11-29
> > > >
> > > > Dear reviewer 6TRe,
> > > >
> > > > We would like to express our sincere gratitude to reviewer 6TRe for acknowledging our work and providing constructive suggestions.
> > > >
> > > > Thanks again for the time and effort in reviewing our work.

---

### Official Review · Reviewer_PKxB · 2024-11-03

**Soundness:** 3
**Presentation:** 3
**Contribution:** 2
**Rating:** 6
**Confidence:** 3

**Summary:**

This paper proposes Pyramid Attention Broadcast (PAB), a method for DiT-based video generation that leverages a pyramid-shaped broadcast design. This design is inspired by two key observations: (1) the attention output differences are minimal in the middle 70% of the diffusion steps, and (2) spatial, temporal, and cross-attention differences decrease in a hierarchical, pyramid-like manner. Based on these insights, PAB sets different broadcast ranges for each attention type. The method is further extended to distributed settings, supporting multi-GPU parallelism (e.g., 8-GPU, 16-GPU) to further reduce generation latency. Experiments on VBench and WebVid using three types of DiT models demonstrate that PAB effectively reduces latency and accelerates video generation.

**Strengths:**

The method’s effectiveness is demonstrated on multi-GPU setups, achieving real-time video generation (e.g., within 2 seconds) on an 8-card H100 configuration. This efficiency is also validated across multiple open-source Video DiT models.

**Weaknesses:**

1. In Figure 4, the quantitative analysis of attention differences uses 30 inference steps based on Open-So ra, but the visualized attention differences cover 50 steps, creating inconsistency. Is this visualization based on Latte? Additionally, it’s unclear whether the attention feature difference plot is derived from an entire dataset analysis or a single video generation process.

2. Some comparisons appear to be unfair. For example, in Figure 7, the original generation using a single device is compared to an 8-device setup, leading to a claimed 10.5x speedup. This comparison may be misleading and should be clarified.

3. Key implementation details should be presented in the main paper. For instance, the most crucial setting, **PAB_246**, which defines the spatial, temporal, and cross-attention broadcast ranges, should be clearly explained rather than shown in supplementary materials.

4. Broadcasting attention is not entirely new. For instance, TGATE, an image generation acceleration approach, also uses iterative caching and reusing of self-attention (SA) for acceleration. This reduces the novelty of the proposed approach.

5. The qualitative results focus primarily on natural, static scenes. In real-world applications, generating videos with complex, dynamic actions, such as those involving people or animals, is more impactful. The qualitative evaluation scope is therefore too limited to showcase the method’s full potential in diverse scenarios.

**Questions:**

The broadcast sequence parallelism in this work relies on DSP, so it’s unclear how much of the latency reduction is directly attributable to PAB alone. In Table 4 or in the main text, it would be helpful to further clarify PAB’s independent contribution to latency improvements.

---

> ### Author Response · Authors · 2024-11-21
> **Response to reviewer PKxB (1/4)**
>
> We sincerely thank the reviewer PKxB for the valuable questions and comments. For the concerns and questions, here are our responses:
>
> **Q1: Is the visualization in Fig 4 based on Latte? Whether the attention feature difference plot is derived from an entire dataset analysis or a single video generation process.**
>
> **A1:** Thanks for the corrrection. The visualization is indeed based on Latte, sorry for the mistake in the paper. We select Latte because its DDIM scheduler samples uniformly, therefore more suitable and clear for visualization of difference.
>
> While the visualization is demonstrated using Latte, we note that our observations generalize well to other models as the results show in our experiments.
>
> Regarding the attention feature difference plot, we conduct our analysis using the complete Vbench dataset to ensure comprehensive and statistically robust results.
>
> **Revise:** In the revision, we correct the caption of Figure 4 (Line 175) from `Visualization of attention differences in Open-Sora.` to `Visualization of attention differences in Latte.`
>
> We will further carefully check all details in the revision. Thanks for pointing out this.
>
>
>
> **Q2: Key implementation details like PAB$_{246}$ should be presented in the main paper.**
>
> **A2:** Thanks for this valuable feedback regarding the presentation of PAB$_{246}$.
>
> We introduce PAB$_{\alpha\beta\gamma}$ in the caption of Table 1 (Line 288), where it's first mentioned. For the convenience, we provide the original caption as follows:
>
> "PAB$_{\alpha\beta\gamma}$ denotes broadcast ranges of spatial ($\alpha$), temporal ($\beta$), and cross ($\gamma$) attentions."
>
> Following the advice, we plan to make the explanations more clear.
>
> **Revise:** In the revision, we clarify this term again in the Table 1 analysis paragraph (Line 266): "PAB$_{\alpha\beta\gamma}$ denotes broadcast ranges of spatial ($\alpha$), temporal ($\beta$), and cross ($\gamma$) attentions."

---

> > ### Author Response · Authors · 2024-11-21
> > **Response to reviewer PKxB (2/4)**
> >
> > **Q3: The qualitative results focus primarily on natural, static scenes.**
> >
> > **A3:** We thank the comment. While the visualized examples focused on natural and static scenes, our method extends well beyond these cases.
> >
> > Our quantitative evaluation in Table 1 (Line 287) on the VBench dataset contains numerous examples of complex, dynamic actions, which partially demonstrate our method's broader effectiveness.
> >
> > We further clarify this as follows:
> >
> > **For model settings**:
> >
> > * We specifically use Open-Sora to generate videos of **16 seconds duration** (the longest duration).
> > * This longer duration purposefully challenges our method with more complex and dynamic scenes.
> > * Open-Sora is the only model used as other models are restricted to fixed short lengths.
> >
> > **For dataset**:
> >
> > * From VBench's comprehensive 16-dimensional evaluation metrics, **we strategically select 7 categories that best assess complex and dynamic scenes**:
> >   * human action
> >   * overall consistency
> >   * imaging quality
> >   * aesthetic quality
> >   * dynamic degree
> >   * motion smoothness
> >   * subject consistency
> > * Total performance scores are averaged based exclusively on these 7 categories to provide evaluation of complex and dynamic capabilities.
> >
> > **Quantitative results**:
> >
> > | method      | human action | overall consistency | imaging quality | aesthetic quality | dynamic degree | motion smoothness | subject consistency | total |
> > | ----------- | ------------ | ------------------- | --------------- | ----------------- | -------------- | ----------------- | ------------------- | ----- |
> > | original    | 92.67        | 73.65               | 61.40           | 56.59             | 21.07          | 96.43             | 90.26               | 74.54 |
> > | PAB$_{246}$ | 91.33        | 73.43               | 60.18           | 56.24             | 19.91          | 97.05             | 90.08               | 73.98 |
> > | PAB$_{357}$ | 89.33        | 72.53               | 58.17           | 54.86             | 19.45          | 96.40             | 88.35               | 72.58 |
> > | PAB$_{579}$ | 88.33        | 72.36               | 57.92           | 54.63             | 18.05          | 96.50             | 88.32               | 72.15 |
> >
> > **Analysis**: As shown in table above, we specifically test our method under the most challenging conditions by using the longest videos and selecting the more complex tasks in the dataset. The results show that PAB performs consistently well, with PAB$_{246}$ achieving comparable performance compared with original method.
> >
> > What's particularly encouraging is that PAB maintains good scores even in the most difficult dimensions we tested, like human action and dynamic degree. This shows that our model stays reliable even under demanding conditions.
> >
> > **Qualitative results:** As shown in Figure 16 (Line 1093), our method demonstrates robust performance in processing dynamic, complex scenes while maintaining high-quality output.
> >
> > Here are some example prompts in qualitative results:
> >
> > * Origami dancers in white paper, 3D render, on white background, studio shot, dancing modern dance.
> > * this is how i make up in the morning.
> > * Sewing machine, old sewing machine working.
> > * a horse running to join a herd of its kind.
> >
> > **Improvement plan**: In the revision, we update both quantitative and qualitative results for PAB with long, complex, and dynamic scenes in appendix B.3 (Line 1015).
> >
> > **Conclusion: Both qualitative and quantitative results show our method is able to handle complex and dynamic videos.**

---

> > > ### Author Response · Authors · 2024-11-21
> > > **Response to reviewer PKxB (3/4)**
> > >
> > > **Q4: Clarify PAB’s independent contribution to latency improvements for multiple GPUs.**
> > >
> > > **A4:** Thanks for your valuable suggestion.
> > >
> > > Our evaluation presented in Table 4 (Line 378) examines the generation across three distinct parallelism methods with and without PAB. The results are compelling: PAB reduces communication volume by 43.3% while achieving speedup of 16.2%, 25.3%, and 29.3% for the three parallel methods.
> > >
> > > To be more clear about how PAB contributes to the speedup in detail. Here we further breakdown the contribution of PAB:
> > >
> > > **Experiments**:
> > >
> > > | method                                                       | latency (s) |
> > > | ------------------------------------------------------------ | ----------- |
> > > | original (1 GPU)                                             | 96.90       |
> > > | DSP (8 GPUs)                                                 | 11.53       |
> > > | DSP + PAB (with computation speedup) (8 GPUs)                | 9.29        |
> > > | DSP + PAB (with computation and communication speedup) (8 GPUs) | 8.85        |
> > >
> > > **Experiment settings**: DSP is the sequence parallel method we use. We follow the settings of Table 4, using Open-Sora to generate 8s of 480p videos. The speed is slightly different from the results in Table 1 (Line 270) because we are using different machines and environments.
> > >
> > > **Analysis**: As shown in the experiment, we evaluate the independent contribution of PAB from computation and communication.
> > >
> > > * With PAB's computation speedup (save computation by attention broadcast), the latency is further reduced by 24% compared with DSP only.
> > > * With PAB's communication speedup (can save all communication costs when temporal attention is skipped), the latency can be further reduced by 5.0% compared with computation speedup only.
> > > * Since we only evaluate based on PAB$_{246}$ (better quality but less speedup), PAB is able to achieve more speedup if using more aggressive strategies.
> > >
> > > **Improvement plan:** In the revision, we update the clarification of PAB’s independent contribution to the appendix B.4 (Line 1110). And in future, we will organize this clarification into Table 4 (Line 378).
> > >
> > > **Conclusion: PAB substantially contributes to both computation and communication.**

---

> > > > ### Author Response · Authors · 2024-11-21
> > > > **Response to reviewer PKxB (4/4)**
> > > >
> > > > **Q5: In Figure 7, the original generation using a single device is compared to an 8-device setup, leading to a claimed 10.5x speedup. This comparison may be misleading and should be clarified.**
> > > >
> > > > **A5**: Thanks for your suggestion. It's indeed misleading for readers that's not familiar with our 8-device setup. We will make it more clear and clarified.
> > > >
> > > > **Revise**: In the revision, we clarify this in caption of Figure 7 (Line 313):  `The multiple GPUs' speedup is compared with single GPU's speed.`
> > > >
> > > > We also add some more experiments about PAB's independent contribution in multiple GPUs senairos in appendix B.4 Table 12 (Line 1123).
> > > >
> > > > Thanks a lot for the suggestion. Looking forward to more discussion!
> > > >
> > > >
> > > >
> > > > **Q6: Broadcasting attention is not entirely new, reducing the novelty of the proposed approach.**
> > > >
> > > > **A6:** Thanks for the comment. PAB is indeed fundamentally a caching method. But we believe the significance of our work extends beyond this technical solution. Our paper presents two crucial empirical findings in video diffusion models:
> > > >
> > > > * There exists redundancy in attention for the middle 70% steps of diffusion.
> > > > * Different attentions have different redundancy degrees. Based on these two key findings, there can be many solutions to improve the efficiency of video models.
> > > >
> > > > Based on these two key findings, there are many methods to improve the efficiency of video generation models. For example:
> > > >
> > > > **Proposal 1:** We can make the model architecture more efficient by changing the number of each attention.
> > > >
> > > > **Motivation**: Now most models use equal amounts of spatial, temporal, and cross attention. However, our research shows that cross attention and temporal attention often do similar things (have high redundancy).
> > > >
> > > > **Method**: Therefore, we suggest changing the number of these attentions:
> > > >
> > > > * Spatial attention: every 1-2 layers (least redundant).
> > > > * Temporal attention: every 2-3 layers (less redundant).
> > > > * Cross attention: every 4-6 layers (most redundant).
> > > >
> > > > **Expected results & analysis**: This adjustment would help the model focus more on the content itself rather than spending too much effort analyzing relationships between elements. This could reduce redundant processing and potentially make the model more efficient.
> > > >
> > > > **Proposal 2:** We can make the diffusion scheduler more efficient by improving sample distribution.
> > > >
> > > > **Motivation**:  Our analysis revealed in Figure 1 that redundancy occurs in the middle 70% of the diffusion process. This insight suggests an opportunity to optimize the noise scheduler strategy by concentrating sampling efforts on the initial and final stages of diffusion, rather than the redundant middle segment.
> > > >
> > > > **Method**: We propose optimizing the noise scheduler by redistributing sampling density:
> > > >
> > > > * Allocate 60-70% of sampling steps to the critical initial (first 20%) and final (last 10%) stages.
> > > > * Reduce sampling ratio to 30-40% in the middle segments where redundancy is highest.
> > > >
> > > > **Expected results & analysis**: This approach could significantly improve both training efficiency and inference performance, and similar ideas have already been adopted by some latest works (https://arxiv.org/pdf/2410.13720, https://arxiv.org/pdf/2405.17403).
> > > >
> > > > **Conclusion:**
> > > >
> > > > * Our proposed method speedup video generation with caching method, **but we view it as an initial step toward a broader goal: inspiring the development of more efficient video generation approaches.**
> > > > * These approaches include **improvements in model architecture** and **noise scheduler optimization** for both during training and inference.
> > > > * We envision our work contributing to **making video generation more practical and accessible for real-world applications**.
> > > >
> > > > We are particularly happy to continue the discussion in the discussion phase!

---

> > > > > ### Author Response · Authors · 2024-11-25
> > > > > **Looking forward to your reply**
> > > > >
> > > > > Dear reviewer PKxB:
> > > > >
> > > > > Thanks so much again for the time and effort in our work. According to the comments and concerns, we conduct the corresponding experiments and further discuss the related points. Additionally, we have added all improvements including 6 detailed experiments in the latest pdf revision.
> > > > >
> > > > > As the rebuttal period is about to close, may I know if our rebuttal addresses the concerns? If there are further concerns or questions, we are welcome to address them. Thanks again for taking the time to review our work and provide insightful comments.

---

> > > > > > ### Comment · Reviewer_PKxB · 2024-11-29
> > > > > > **Response**
> > > > > >
> > > > > > Dear Authors,
> > > > > >
> > > > > > Thank you for the detailed response. Your rebuttal has well addressed my concerns. I will keep my rating of acceptance.

---

> > > > > > > ### Author Response · Authors · 2024-11-29
> > > > > > >
> > > > > > > Dear reviewer PKxB,
> > > > > > >
> > > > > > > We would like to express our sincere gratitude to reviewer PKxB for acknowledging our work and providing constructive suggestions.
> > > > > > >
> > > > > > > Thanks again for the time and effort in reviewing our work.

---

### Official Review · Reviewer_R3bt · 2024-11-04

**Soundness:** 3
**Presentation:** 3
**Contribution:** 2
**Rating:** 6
**Confidence:** 4

**Summary:**

The paper propose one technique, namely pyramid attention broadcast (PAB), which is a training-free for speed-up DiT-based video generation.
The idea is simple, which broadcast the attention results for the spatial, temporal, cross attention, which consumes more computation in video DiTs than that in CNN approaches.
The proposed PAB works well in OpenSora, Open-Sora-Plan, and Latte, demonstrating the effectiveness of the proposed method.

**Strengths:**

- The motivation is clear, which relies on one training-free method to speedup the video generation (inference).
- The attention redundancy in video DiTs are extensively studied, such as the attention cost, the attention patterns, the attention similarity and diversity.
- With the extensive studies on attention redundancy, the authors proposed the pyramid attention broadcast. Moreover, different broadcast ranges for each attention are tailored based on the rate of change and the stability of each attention type.
- Experiments and ablation studies are conducted to demonstrate the effectiveness and efficiency of the proposed PAB metric.

**Weaknesses:**

- The proposed PAB is simple and effective. However, the optimal solution for PAB, such as how to determine the optimal broadcast ranges are not specified. According to the Figure. 8, the larger broadcast range, the low latency and low video generation quality. Such studies seed to be obvious to the authors. For the specific video generation model, such as open-sora, open-sora-plan, latte, how can we set the broadcast range. Please provide more discussions.
- The authors claimed that the broadcast the attention outputs other than attention scores. In my opinion, the effect seems to be the same or very similar for broadcasting the attention scores outputs. Please provide more explanation.
- One important questions is that the authors reveal that the attention outputs (spatial, temporal, cross attention) are redundancy. As such, they can broadcast the attention outputs with one training-free to speedup the video generation. In my opinion, such studies also means that the existing video generation model, such as open-sora, open-sora-plan, latte, learns a lot of redundancy information, which results in the redundancy attention outputs. Therefore, can PAB helps to learn effectively of the video generation model. Please make more discussions.

**Questions:**

Please check the detailed comments in the weaknesses part.

---

> ### Author Response · Authors · 2024-11-21
> **Response to reviewer R3bt (1/3)**
>
> We sincerely thank the reviewer R3bt for the valuable questions and comments. For the concerns and questions, here are our responses:
>
> **Q1: How to determine the optimal broadcast ranges?**
>
> **A1:** Thanks for the comment. There are only two hyperparameters in PAB: 1) broadcast range. 2) the beginning and end steps of the broadcast. And both of them are easy to find the best hyper parameters.
>
> 1. **Broadcast range**
>
> **Experiment settings**: We conduct the following experiment to find out how to determine the best broadcast for all methods, where PAB$_{\alpha\beta\gamma}$ denotes broadcast ranges of spatial ($\alpha$), temporal ($\beta$), and cross ($\gamma$) attentions. The italicized results mean significant performance degradation.
>
> | method         | original | PAB$_{222}$ | PAB$_{235}$ | PAB$_{246}$ | PAB$_{257}$  | PAB$_{333}$  |
> | -------------- | -------- | ----------- | ----------- | ----------- | ------------ | ------------ |
> | Open-Sora      | 79.22    | 78.63       | 78.59       | **78.51**   | *77.88* | *78.07* |
> | Open-Sora-Plan | 80.39    | 80.33       | 80.33       | **80.30**   | *79.54* | *78.83* |
> | Latte          | 77.40    | 76.55       | **76.32**   | 76.25       | *76.02* | *75.86* |
>
> **Analysis**: As shown in the table, we can discover the following findings:
>
> * When we increase broadcast range from PAB$\_{222}$ to PAB$\_{246}$ or PAB$\_{235}$ , there is not much performance degradation, as the redundancy degree of temporal and cross attention is larger. **So broadcast range should follow: cross range > temporal range > spatial range**.
> * When extending to PAB$\_{257}$ or PAB$\_{333}$, there will be severe performance degradation.
> * Although different models have different schedulers, timesteps, and training strategies, their best broadcast ranges are similar. **The optimal range is either PAB$\_{246}$ or PAB$\_{235}$** considering both effect and efficiency.
>
> 2. **The beginning and end steps of broadcast**
>
> **Experiment settings**: We conduct experiments to find out how to determine the best start and end steps for all methods, where $[\cdot, \cdot]$ denotes the end and start diffusion step of broadcast. The more steps it covers, the more speedup PAB can achieve. The italicized results mean significant performance degradation.
>
> | method         | original | [200, 700] | [150, 750] | [100, 800] | [100, 850] | [50, 950]    |
> | -------------- | -------- | ---------- | ---------- | ---------- | ---------- | ------------ |
> | Open-Sora      | 79.22    | 78.65      | 78.77      | **78.51**  | 78.20      | *77.21* |
> | Open-Sora-Plan | 80.39    | 80.30      | 80.22      | 80.28      | **80.30**  | *78.13* |
> | Latte          | 77.40    | 76.54      | 76.53      | **76.32**  | 75.90      | *75.66* |
>
> **Analysis**: As shown in the table, we can discover the following findings:
>
> * Reducing broadcast steps to $[150,750]$ or $[200,700]$ does not increase worthy performance, but leading to less speedup. However, increasing broadcast steps beyond $[50,950]$ leads to significant degradation.
> * Similarly, although different models have different schedulers, timesteps and training strategies, their best strategy is similar. **The optimal strategy is either $[100,800]$ or $[100,850]$**.
>
> **Conclusion: PAB is easy to find optimal hyperparameters.** It can consistently achieve good results for all three methods by setting:
>
> * **Broadcast range to PAB$\_{246}$ or PAB$\_{235}$**.
> * **Broadcast beginning and end steps to $[100,800]$ or $[100,850]$**.

---

> > ### Author Response · Authors · 2024-11-21
> > **Response to reviewer R3bt (2/3)**
> >
> > **Q2: Why do the authors claim that broadcast the attention outputs other than attention scores?**
> >
> > **A2:** Thanks for the comment. Our choice to broadcast attention outputs rather than attention scores is supported by our experiments considering both video quality and speed.
> >
> > As demonstrated in Table 3 (Line 324), while both approaches yield comparable video quality, broadcasting attention outputs significantly outperform attention scores in terms of computational efficiency.
> >
> > For the convenience, we show Table 3 (Line 324) as follows:
> >
> > | broadcast object  | Vbench (%) | latency (s) |
> > | ----------------- | ---------- | ----------- |
> > | original          | 79.22      | 26.54       |
> > | attention scores  | 78.53      | 29.12       |
> > | attention outputs | 78.51      | 19.87       |
> >
> > This performance gap comes from the following advantages:
> >
> > * Broadcasting attention outputs enables us to bypass all intermediate computations within the attention module (including layer normalization, positional embedding, and qkvo projections) while maintaining compatibility with efficient attention kernels such as FlashAttention (we enable FlashAttention in all experiments by default to be closer to real-world usage).
> > * But broadcasting attention scores still requires partial computation in the attention module (e.g., attention calculation and linear projection). Its performance may even **degrade below baseline due to incompatibility with FlashAttention**.
> >
> > To be more clear, here are the workflows in attention module for different broadcast strategies:
> >
> > * **original**:
> >
> >   $x \to q,k,v=proj(x) \to q,k=pos\\_emb(layer\\_norm(q,k)) \to o=attn(q,k,v) \to o=proj(o)$
> >
> > * **attention score** (cannot use FlashAttention because we need attention score explicitly):
> >
> >   $x \to \cancel{q,k},v=proj(x) \to \cancel{q,k=pos\\_emb(layer\\_norm(q,k))} \to o=attn(broadcast\\_score,v) \to o=proj(o)$
> >
> >   simplified: $x \to v=proj(x) \to o=attn(broadcast\\_score,v) \to o=proj(o)$
> >
> > * **attention outputs:**
> >
> >   $x \to \cancel{q,k,v=proj(x)} \to \cancel{q,k=pos\\_emb(layer\\_norm(q,k))} \to \cancel{o=attn(q,k,v)} \to \cancel{o=proj(o)} \to o=broadcast\\_outputs$
> >
> >   simplified: $o=broadcast\\_outputs$
> >
> > **Improvement plan:** In the revision, we update the above workflow of attention module of these three methods in the appendix B.6 (Line 1162).
> >
> > **Conclusions**:
> >
> > * Both strategies’ video quality is similar.
> > * We choose **broadcast attention outputs** for **better efficiency**.

---

> > > ### Author Response · Authors · 2024-11-21
> > > **Response to reviewer R3bt (3/3)**
> > >
> > > **Q3: Can PAB help to learn effectively of the video generation model?**
> > >
> > > **A3:** Thanks for the comment. This is an interesting question! We briefly mentioned this in future work (Line 535). Definitely, we're willing to explain it here. We introduce two promising directions for leveraging PAB to enhance model training effectiveness:
> > >
> > > **Proposal 1:** We can make the model architecture more efficient by changing the number of each attention.
> > >
> > > **Motivation**: Now most models use equal amounts of spatial, temporal, and cross attention. However, our research shows that cross attention and temporal attention often do similar things (have high redundancy).
> > >
> > > **Method**: Therefore, we suggest changing the number of these attentions:
> > >
> > > * Spatial attention: every 1-2 layers (least redundant).
> > > * Temporal attention: every 2-3 layers (less redundant).
> > > * Cross attention: every 4-6 layers (most redundant).
> > >
> > > **Expected results & analysis**: This adjustment would help the model focus more on the content itself rather than spending too much effort analyzing relationships between elements. This could reduce redundant processing and potentially make the model more efficient.
> > >
> > > **Proposal 2:** We can make the diffusion scheduler more efficient by improving sample distribution.
> > >
> > > **Motivation**:  Our analysis revealed in Figure 1 (Line 81) that redundancy occurs in the middle 70% of the diffusion process. This insight suggests an opportunity to optimize the noise scheduler strategy by concentrating sampling efforts on the initial and final stages of diffusion, rather than the redundant middle segment.
> > >
> > > **Method**: We propose optimizing the noise scheduler by redistributing sampling density:
> > >
> > > * Allocate 60-70% of sampling steps to the critical initial (first 20%) and final (last 10%) stages.
> > > * Reduce sampling ratio to 30-40% in the middle segments where redundancy is highest.
> > >
> > > **Expected results & analysis**: This approach could significantly improve both training efficiency and inference performance, and similar ideas have already been adopted by some latest works (https://arxiv.org/pdf/2410.13720, https://arxiv.org/pdf/2405.17403).
> > >
> > > **Conclusion**: To improve the efficiency of video generation model, we can **1) adjust the number of each attention. 2) focusing diffusion steps on more important stages.**

---

> > > > ### Author Response · Authors · 2024-11-25
> > > > **Looking forward to your reply**
> > > >
> > > > Dear reviewer R3bt:
> > > >
> > > > Thanks so much again for the time and effort in our work. According to the comments and concerns, we conduct the corresponding experiments and further discuss the related points. Additionally, we have added all improvements including 6 detailed experiments in the latest pdf revision.
> > > >
> > > > As the rebuttal period is about to close, may I know if our rebuttal addresses the concerns? If there are further concerns or questions, we are welcome to address them. Thanks again for taking the time to review our work and provide insightful comments.

---

> ### Comment · Reviewer_R3bt · 2024-11-26
> **To authors**
>
> The authors have done additional experiments to answer my comments, which are interesting and insightful, such as the "broadcast the attention outputs vs. attention scores", and an efficient video generation model.
>
> I am standing on my previous rating.

---

> > ### Author Response · Authors · 2024-11-26
> >
> > Dear reviewer R3bt,
> >
> > We would like to express our sincere gratitude to reviewer R3bt for acknowledging our work and providing constructive suggestions.
> >
> > Thanks again for the time and effort in reviewing our work.

---

### Author Response · Authors · 2024-11-21
**Update our work's pdf with additional experiments and refining some minors**

Dear ACs and reviewers,

We sincerely thank you for the time and effort in our work.

Additional experiments were updated, and we went through our work to fix some minors. All changes are highlighted in red.

We will continue to refine our work.

Thanks,

Authors of submission 4140

---

### Author Response · Authors · 2024-11-25
**Update pdf with ALL additional experiments and paragraphs**

Dear ACs and reviewers,

We sincerely thank you for the time and effort in our work.

We have further polished our work, and updated ALL additional experiments and paragraphs (highlight in red).

If you still have any questions, feel free to ask us! Looking forward to some discussion!

Thanks,

Authors of submission 4140

---

### Meta-Review · Area_Chair_iv7i · 2024-12-18

**Metareview:**

(a) Scientific Claims and Findings

The paper identifies significant redundancy in attention modules across diffusion steps in DiT-based video generation models. Based on this observation, the authors propose Pyramid Attention Broadcast (PAB), a method that reuses attention outputs across steps. This approach reduces computational overhead without sacrificing video quality.

(b) Strengths
* Clear Motivation and Analysis: The paper provides a clear motivation for addressing redundancy in attention modules and includes extensive empirical studies on attention redundancy within DiT models.
* Clear Speedup: The paper shows convincing evidence of PAB’s ability to reduce inference time.
* Extensive Evaluation: The authors conduct thorough ablation studies and evaluate the proposed method across multiple models to validate the effectiveness of the method.

(c) Weaknesses
* Lack of Clarity on Approach Details: The paper does not provide sufficient implementation details and does not explain the process for determining hyperparameters, such as the optimal broadcast range and diffusion steps.
* Fairness of Comparisons: The claim of a 10x speedup is not valid as it compares between single-GPU and multi-GPU setups.
* Novelty Concerns: The core idea of caching and reusing attention outputs has been explored in prior works, such as T-GATE and DeepCache.
* Limited Qualitative Results: The qualitative results are limited to relatively static and simple domains, raising concerns about the method’s generalizability to more dynamic and complex scenarios.

(d) Reasons for Accept / Reject

Despite its weaknesses, the paper is well-presented, and the authors provide a thorough empirical study to support their claims. The speedup introduced by PAB is well justified by empirical results, demonstrating a meaningful reduction in computation time. However, the contribution is somewhat constrained due to two key factors: the actual speedup achieved (approximately 30%) is far from authors' original 10x claim, and the novelty of the PAB is limited given it's similarity with existing approaches.

**Additional Comments On Reviewer Discussion:**

During the rebuttal, the authors provided detailed responses and additional experiments that addressed many of the reviewers' concerns. They clarified implementation details, including the process for tuning hyperparameters, and presented a new analysis of the multi-GPU speedup, though the original 10x claim remains overstated. The authors also emphasized new empirical findings on attention redundancy as a key contribution beyond the caching method. Overall, the reviewers were satisfied with the clarifications and updates, resulting in either maintained or increased ratings. However, concerns about the validity of the 10x speedup claim and the generalizability to more dynamic and longer videos remain unresolved.

---

### Decision · Program_Chairs · 2025-01-22

Accept (Poster)